# Deubiquitinases: Modulators of Different Types of Regulated Cell Death

**DOI:** 10.3390/ijms22094352

**Published:** 2021-04-21

**Authors:** Choong-Sil Lee, Seungyeon Kim, Gyuho Hwang, Jaewhan Song

**Affiliations:** 1Integrated OMICS for Biomedical Science, World Class University, Yonsei University, Seoul 120-749, Korea; lcs07052@yonsei.ac.kr; 2Department of Biochemistry, College of Life Science and Biotechnology, Yonsei University, Seoul 120-749, Korea; cherry0313@yonsei.ac.kr (S.K.); hgh1994@yonsei.ac.kr (G.H.)

**Keywords:** deubiquitinase, apoptosis, necroptosis, pyroptosis, ferroptosis, paraptosis, autophagic cell death, physiology

## Abstract

The mechanisms and physiological implications of regulated cell death (RCD) have been extensively studied. Among the regulatory mechanisms of RCD, ubiquitination and deubiquitination enable post-translational regulation of signaling by modulating substrate degradation and signal transduction. Deubiquitinases (DUBs) are involved in diverse molecular pathways of RCD. Some DUBs modulate multiple modalities of RCD by regulating various substrates and are powerful regulators of cell fate. However, the therapeutic targeting of DUB is limited, as the physiological consequences of modulating DUBs cannot be predicted. In this review, the mechanisms of DUBs that regulate multiple types of RCD are summarized. This comprehensive summary aims to improve our understanding of the complex DUB/RCD regulatory axis comprising various molecular mechanisms for diverse physiological processes. Additionally, this review will enable the understanding of the advantages of therapeutic targeting of DUBs and developing strategies to overcome the side effects associated with the therapeutic applications of DUB modulators.

## 1. Introduction

### 1.1. Regulated Cell Death (RCD)

Regulated cell death progresses through defined molecular cascades that cells can modulate to determine cell fate. Previous studies have examined the molecular characteristics and physiological significance of different types of RCD [1,2,3].

This review focuses on deubiquitinases (DUBs), which are regulatory factors of RCD, shared among different types of RCD. DUBs play critical roles in intrinsic apoptosis, a widely studied type of RCD. Additionally, this review focuses on the role of DUBs in extrinsic apoptosis, necroptosis, pyroptosis, paraptosis, ferroptosis, and autophagic cell death. Studies on the roles of DUBs in other types of RCD are limited; hence, the correlation between DUBs and other types of RCD has been excluded from this review.

### 1.2. Apoptosis

Apoptosis, a type of RCD, is characterized by cell shrinkage, membrane blebbing, and DNA fragmentation. Generally, apoptosis is considered a “non-inflammatory cell death”, as the cell debris is engulfed by the immune cells. Extrinsic ligands and cellular stress trigger apoptosis through the activation of caspases. Intrinsic apoptosis is induced by various cellular stresses, such as growth factor depletion, DNA damage, endoplasmic reticulum (ER) stress, increased reactive oxygen species (ROS) generation, replication stress, microtubular alterations, or mitotic defects. These stimuli promote mitochondrial outer membrane permeabilization (MOMP), which is the formation of pores in the mitochondrial outer membrane. Among the BCL-2 family proteins, BAX and BAK mediate MOMP, whereas BCL-XL, MCL1, BCL-2, and BFL-1 inhibited the activities of BAX and BAK. MOMP activation results in the release of mitochondrial components, such as CYCS and SMAC. APAF1 and procaspase-9 assemble to form an apoptosome, which activates the initiator caspase (caspase-9). Activated caspase-9 cleaves caspase-3 and caspase-7, which subsequently activate downstream proteins [1,2,3].

By cleaving the inhibitor of CAD (ICAD), caspase-3 and caspase-7 activate caspase-activated DNase (CAD) to degrade DNA. Acinus, which is activated by caspase-3, promotes chromatin condensation. Caspase-6 cleaves and inactivates lamins, resulting in nuclear fragmentation. To induce apoptotic morphology, activated caspases cleave proteins that regulate cell structure, such as gelsolin, α-fodrin, focal adhesion kinase, PAK2, and actin [4]. Activated caspase-3 promotes membrane localization of phosphatidylserine by activating phosphatidylserine efflux and deactivating phosphatidylserine influx (Figure 1a) [1,2,3].

The interaction between extracellular ligands and their cognate ligands activates extrinsic apoptosis. The binding of ligands, such as tumor necrosis factor-alpha (TNFα), Fas ligand (FasL), and TNF-related apoptosis-inducing ligand (TRAIL) to death receptors, such as tumor necrosis factor receptor (TNFR), Fas cell surface death receptor (Fas; also known as CD95), and TRAIL receptor (death receptor 4 (DR4) and 5 (DR5)), respectively, results in the formation of a death-inducing signaling complex (DISC). In DISC, caspase-8 is activated by binding to FADD. The activation of caspase-8 is inhibited by the binding of cellular FLICE-inhibitory protein (c-FLIP) (both long and short forms) to DISC. The extrinsic apoptosis pathway converges with the downstream cascades of the intrinsic apoptosis pathway after caspase-8 activates caspase-3 (Figure 1b) [1,2,3].

Anoikis, which is induced by the detachment of cells from the extracellular matrix, is mediated by both intrinsic and extrinsic apoptosis-related factors. Hence, anoikis is critical for the elimination of non-adherent cells, such as tumor cells [5].

Dependence receptors, which monitor the levels of their ligands in the surrounding environment, can induce caspase-dependent apoptosis when their ligands do not reach the threshold concentration. DCC, UNC5B (receptor for netrin 1), PTCH1 (receptor for sonic hedgehog), and NTRK3 (receptor for neurotrophin), which are mediators of dependence receptor signaling, induce apoptosis through the activation of caspases, p53, or E2F1 [1,6].

### 1.3. Necroptosis

Extracellular and intracellular insults activate caspase-independent necroptosis through sequential phosphorylation and activation of RIPK3 and MLKL. Activated MLKL undergoes oligomerization and induces plasma membrane rupture and cell swelling, which exposes damage-associated molecular patterns (DAMPs) or pathogen-associated molecular patterns (PAMPs) in the microenvironment. This results in a proinflammatory response in the surrounding microenvironment, which is distinct from the inflammatory response activated during apoptotic cell death in which cellular materials are trapped in apoptotic bodies [2]. Various stimuli can initiate necroptosis, including the activation of death receptors, pathogen recognition receptors, and endogenous nucleic acid sensors. These signals are transduced to MLKL through interactions with proteins containing RHIM. TNFR-induced necroptosis is activated through the RHIM-mediated interaction between RIPK1 and RIPK3, promoting the recruitment of MLKL. After recognizing pathogenic invasion, Toll-like receptors induce necroptosis by promoting RHIM-mediated interaction between TRIF and RIPK3, which subsequently activates MLKL. Z-DNA-binding protein 1 (ZBP1), a sensor for cytosolic double-stranded nucleic acids, activates RIPK3 and MLKL (Figure 1c) [1,2,3].

### 1.4. Pyroptosis

Pyroptosis, an inflammatory cell death, is induced by DAMPs, PAMPs, lipopolysaccharide (LPS), TAK1 inhibition, and death receptors. Exposure to DAMP/PAMP recruits the inflammasome, in which caspase-1 is activated. Activated caspase-1 promotes IL-1β and IL-18 maturation and secretion and cleaves gasdermin D (GSDMD). The N-terminal fragment of GSDMD forms pores in the plasma membrane and induces a pyroptosis phenotype. Alternatively, cytosolic LPS binds and activates caspase-4/5 (caspase-11 in mice). The activation of caspase-4/5 promotes the formation of the inflammasome and thus converges with the activation of caspase-1. Death receptor-activated caspase-3 cleaves GSDME. The N-terminal fragment of GSDME induces pyroptosis (Figure 1d). The characteristics of pyroptosis include both apoptotic-like and necrosis-like phenotypes, including chromatin condensation, DNA fragmentation, pore formation, and cell swelling. Previous studies have reported the role of pyroptosis in various physiological processes, including infection, autoimmune diseases, diabetic kidney disease, neurodegenerative diseases, and cancer [7,8,9,10].

The critical role of deubiquitination in pyroptotic cell death was identified using G5, a DUB inhibitor. Treatment with G5 suppresses the pyroptosis cascade and upregulates the ubiquitination of the NOD-like receptor pyrin domain-containing protein 3 (NLRP3) [11].

### 1.5. Ferroptosis

Ferroptosis, which was first identified in 2012, is characterized by iron accumulation, lipid peroxidation, and shrunken mitochondria with increased membrane density and decreased the number of cristae. Previous studies have reported that ferroptosis is involved in various diseases, including cancer, ischemic reperfusion, fibrotic diseases, and autoimmune diseases. Ferroptosis is induced by multiple molecular cascades, all of which lead to the accumulation of phospholipid hydroperoxides. System Xc−, which comprises the SLC7A11/SLC3A2 heterodimer, sustains reductive cellular capacity by importing cystine into the cells and exporting glutamate. The transported cystine is reduced to cysteine, which is utilized to synthesize glutathione (GSH). Glutathione peroxidases (GPXs) utilize GSH to downregulate ROS levels (Figure 1e). Inhibition of system Xc−/GPX was the first identified stimulus for inducing ferroptosis. To date, all DUBs regulating ferroptosis are involved in this pathway. The role of DUBs in other ferroptosis cascades mediated by p53, lipid metabolism, iron metabolism, and p62 is an active area of research [12,13].

The correlation between ferroptosis and DUBs was examined using a general DUB inhibitor. The broad-spectrum DUB inhibitor PdPT downregulates the protein levels of GPX4 and upregulates GPX4 ubiquitination. PdPT treatment induces both apoptosis and ferroptosis. DUB inhibitor-mediated induction of GPX4 proteasomal degradation has been suggested as a potential anticancer therapy because cancer cell lines and primary cancer cells are sensitive to ferroptosis [14].

### 1.6. Paraptosis

Paraptosis or paraptosis-like cell death is characterized by ER dilation, mitochondrial swelling accompanied by ER/mitochondrial stress, and disruption of proteostasis and ion/redox homeostasis. The molecular cascade that directly mediates paraptosis has not yet been identified. However, c-JUN N-terminal kinases (JNK) 1, MEK2, and the inhibitory protein AIP1/Alix are reported to be involved in paraptosis (Figure 1f) [15]. The correlation between paraptosis and DUBs has been reported, although the molecular mechanism underlying paraptosis has not been elucidated.

One study demonstrated a general correlation between DUBs and paraptosis using chemical compounds. Treatment with the hinokitiol copper complex (HK-Cu), a broad proteasomal DUB inhibitor, increases the accumulation of ubiquitinated proteins and induces paraptosis-like cell death. Inhibitors of apoptosis, autophagy, necroptosis, ferroptosis and parthanatos do not suppress HK-Cu-induced cell death. However, elimination of ROS suppresses HK-Cu-induced cell death [16].

### 1.7. Autophagy-Dependent Cell Death

Autophagy-dependent cell death is dependent on autophagy and is not accompanied by apoptosis or necroptosis. Recent studies have reported that the upregulation of autophagy, mitophagy, and autosis promotes cell death. The upregulation of autophagy results in the formation of vacuoles and the depletion of cellular organelles. Deregulation of mitophagy is characterized by ATP depletion. Autosis, which is mediated by Na^+^/K^+^-ATPase, is characterized by membrane rupture and organelle ballooning (Figure 1g). The physiological roles and molecular regulation of autophagic cell death have not been completely elucidated. However, autophagy-related cell death has been reported to be critical for developing embryos in various animal models, including *Drosophila* [1,17].

## 2. DUBs

Ubiquitination involves the conjugation of a 76-amino acid protein called ubiquitin (Ub) to substrate proteins. The ubiquitination at lysine and methionine-1 residues is considered to be canonical ubiquitination. Noncanonical ubiquitination refers to ubiquitination at serine, threonine, and cysteine residues. The following three enzymes catalyze ubiquitination: E1, E2, and E3. E1 catalyzes the ATP-dependent activation of Ub. Activated Ub forms a thioester bond with E2, which transfers Ub to the substrate along with E3 ligase. The substrates are conjugated with the monomers or polymers of Ub. Polyubiquitination is classified based on the linkage between Ub monomers. Both canonical and noncanonical ubiquitination have critical impacts on cellular functions, including protein degradation and signaling cascades through ubiquitination patterns, including mono-ubiquitination, poly-ubiquitination with variations of linkage types such as M1-polyubiquitination and K48-polyubiquitination [18,19,20,21,22].

DUBs modulate the stability and signaling activity of substrates by cleaving the ubiquitin conjugates on the substrates (Figure 2a). Based on evolutionary conservation, DUBs are classified into USP, UCH, OTU, MJD, JAMM, MINDY, and ZUP1 subfamilies (Figure 2b,c). DUBs have many substrates and are involved in diverse cellular functions, such as gene expression, DNA repair, cell cycle progression, differentiation, signaling cascades, protein quality control, and metabolism. Thus, DUBs are associated with physiological and pathological processes, such as cancer, immune disorders, infectious diseases, neuronal diseases, metabolism, and vascular pathology. Various inhibitors of DUBs have been developed for the clinical treatment of human pathologies [18,19,23,24,25].

In particular, DUBs regulate the molecular cascades of RCD and determine cell survival and death. DUBs are critical mediators of the pathological roles of RCD, such as infection, tissue injury, degenerative diseases, cancer, development, and tissue homeostasis [1,26,27]. In this review, we focus on the roles of DUBs in regulating diverse types of RCD and physiological processes.

## 3. DUBs Regulating Diverse RCD

Several DUBs have been revealed to regulate multiple types of RCD through their diverse substrates and molecular pathways. Downstream factors mediating the regulation of RCD by DUBs include factors that directly modulate RCDs, such as BAX, RIPs, and c-FLIP, and regulators of other cellular functions, such as histone, AKT, and p62. These diverse downstream cascades of DUBs result in complex regulatory effects of DUBs on RCD. First, each DUB can modulate different types of RCD. Second, each DUB can either promote or suppress the same type of RCD, depending on the downstream signaling. For example, USP7 promotes intrinsic apoptosis through p53, SUV39H1, and PLK1 and suppresses intrinsic apoptosis through MDM2 and Maf and suppresses the ER stress response. In addition, USP7 promotes extrinsic apoptosis by promoting RIPK1 activity and ferroptosis by suppressing SCL7A11 expression (Figure 3a) [28,29,30,31,32,33,34,35,36,37,38,39,40,41,42,43,44]. BAP1 provides another example of the complex regulation of RCD through diverse mediators. BAP1 promotes or suppresses intrinsic apoptosis by regulating ER function, survivin expression, histone ubiquitination, and 14-3-3 activity. BAP1 further modulates extrinsic apoptosis and ferroptosis by regulating the transcription of DR4/5 and SCL7A11 (Figure 3b) [45,46,47,48,49,50,51,52,53,54].

Consequently, of these divergent molecular regulations, DUBs are involved in diverse physiological and pathological processes associated with RCD. For example, UCHL1 modulates several physiological and pathological processes, including malignant diseases, tissue homeostasis, tissue injury, metabolic diseases, and degenerative diseases, by regulating RCD. UCHL1 suppresses diverse types of cancer by enhancing intrinsic apoptosis. The positive UCHL1/apoptosis axis is required for spermatogenesis and is involved in suppressing cardiac hypertrophy. In contrast, the deregulated UCHL1/apoptosis axis aggravates hypoxic brain injury. By suppressing intrinsic apoptosis, UCHL1 ameliorates type 2 diabetes and Alzheimer’s disease. Conversely, UCHL1 promotes high-glucose-induced extrinsic apoptosis and necroptosis, which promote diabetic nephropathy (Figure 4a) [55,56,57,58,59,60,61,62,63,64,65,66,67,68,69,70].

In malignant diseases, DUBs can aggravate or ameliorate cancers, depending on their downstream RCD. For example, UCHL5/USP14 suppresses apoptosis and autophagic cell death to promote cancer. Several types of cancers are sensitized to cell death by depleting or pharmacological inhibition of UCHL5/USP14 (Figure 4b). In contrast, BAP1 promotes or suppresses malignant diseases via RCD. Apoptosis and ferroptosis mediate the tumor-suppressive function of BAP1, whereas BAP1 suppresses apoptosis to protect certain types of cancer cells (Figure 4c) [45,46,47,48,49,50,51,52,53,54].

These examples show the complicated nature of the DUB/RCD/physiology regulatory axis. As described below, 19 DUBs have been found to regulate multiple types of RCD. This is summarized in Table 1. The following sections describe the mechanisms through which each DUB modulates several types of RCD and their physiological functions through various molecular mechanisms in detail.

### 3.1. USP5

Although USP5 preferentially cleaves free ubiquitin chains that are not anchored to substrates, it also deubiquitinates diverse proteins. Several cellular physiological and pathological functions, including development, heat-shock-induced stress response, immune response, cancer, and neuropathology, are regulated by the USP5-mediated deubiquitination of various substrates [192]. USP5 suppresses both intrinsic and extrinsic apoptosis by modulating c-Maf, p14/p53, JNK, caspases, miRNA, RNA-binding protein, and DNA damage response. Interestingly, some tumor cells activate mechanisms that suppress USP5 inhibition-mediated apoptosis. This suggests the molecular mechanisms by which cancer cells develop resistance to USP5 inhibition. However, the USP5/RCD axis also has been reported to modulate non-malignant tissue functions, including tissue development. Therefore, cancer therapy involving cell death targeting through USP5 can evoke unexpected cellular and physiological side effects.

USP5 deubiquitinates and stabilizes c-Maf, a transcription factor related to tumor and immune cell differentiation, and suppresses apoptosis in multiple myeloma cells. The depletion of USP5 promotes apoptosis, which is suppressed upon overexpression of c-Maf [71]. Short hairpin RNA (shRNA) barcode screening revealed that USP5 is a therapeutic target for pancreatic ductal adenocarcinoma (PDAC). The expression of USP5 is upregulated in PDAC tissue but not in chronic pancreatitis and healthy pancreas. The depletion of USP5 induces DNA damage and apoptosis in PDAC cells [72]. In addition to intrinsic apoptosis, USP5 suppresses extrinsic apoptosis in cancer cell lines. TRAIL and FasL suppress the activity of USP5 through caspase-mediated cleavage of USP5 in TRAIL-sensitive cells, but not in TRAIL-insensitive cells. G9(+)-mediated inhibition of USP5 abolished acquired TRAIL resistance and enhanced apoptosis in cancer cell lines [73]. In addition to its roles in tumors, USP5 suppresses apoptosis in cells that differentiate into the photoreceptor in *Drosophila*. The depletion of USP5 induces developmental defects in the eye through the dysregulation of apoptosis and the JNK pathway [74].

The regulation of USP5 also has been reported to affect apoptosis and physiological functions of USP5. USP5 is reported to protect hepatocellular carcinoma (HCC) against apoptosis by suppressing p14ARF/p53 signaling in HCC tissues and cell lines. Hpn, a protein from *Helicobacter pylori*, exhibits growth-inhibitory effects against HCC by suppressing the expression of USP5, consequently activating the p53-p14ARF pathway and inducing apoptosis [75,76]. miR-125a suppresses USP5 expression and exhibits antitumor activity by suppressing cancer cell proliferation and inducing apoptosis. The ectopic expression of USP5 downregulates miR-125a mimic-induced apoptosis in multiple myeloma cells [77]. USP5 inhibition-mediated apoptosis is suppressed by its downstream RNA-binding proteins in glioma cell lines. This suggests that cancer cells can activate mechanisms to overcome USP5 inhibition-mediated apoptosis. The depletion of USP5 destabilizes the hnRNPA1 protein, which is rescued upon treatment with MG132. The downregulation of hnRNPA1 increases the levels of SF2/ASF1, which suppresses bortezomib-induced apoptosis. Thus, inhibition of both USP5 and SF2/ASF1 significantly increases the apoptotic response to bortezomib [78].

### 3.2. USP7 (HAUSP)

USP7 contains a catalytic USP domain between the N-terminal poly-Q/TRAF-like domain and the C-terminal Ub-like domain. Previous studies have examined the role of USP7, a representative DUB of p53, in human pathological processes, including cancer, neuronal disorders, and metabolic disorders [193]. The molecular mechanism by which USP7 inhibitors induce cell death involves p53-mediated intrinsic apoptosis. However, USP7 regulates RCD by modulating other targets, such as ER stress/ROS, Tip60, PLK1, Maf, DNA repair, and epigenetic regulators in various physiological models (Figure 3a). Thus, USP7 is a highly multifunctional regulator of RCD. Therefore, the role of USP7 in physiological processes and its effect on the downstream pathways must be analyzed to evaluate the therapeutic potential of USP7.

USP7 regulates p53-mediated apoptosis by directly stabilizing p53 and indirectly by regulating the transcriptional activity of p53. USP7 stabilizes p53 directly through deubiquitination. In contrast, USP7 stabilizes MDM2 to suppress p53 stability. The final consequence is dependent on the experimental and physiological conditions [30,34,36,194,195]. In addition, USP7 is recruited to p53 target gene promoters, where it sustains SUV39H1 and H3K9me3, which results in blocking the transcription of p53 target genes. Thus, USP7 inhibits p53-mediated apoptosis by regulating chromatin modifications [40]. The therapeutic applications of pharmacological inhibitors of USP7 in cancer have been further examined. DUB inhibitors induce p53-mediated apoptosis in neuroblastoma and colorectal cancer cell lines and BAX-mediated apoptosis in HCC cell lines [39]. P5091, a USP7 inhibitor, induces apoptosis and necrosis, which are accompanied by enhanced autophagy. The p53 WT ovarian cancer cell line is more sensitive to P5091 than the p53 mutant cell line. This indicates that USP7 inhibitors deregulate cellular metabolic processes [42]. In addition to their roles in cancer, P5091 and P22077 induce p53-dependent apoptosis in senescent cells. P5091 alleviates the phenotypes of doxorubicin-induced senescence in mice by eliminating senescent cells and mitigating senescence-associated secretory phenotype (SASP) [33].

The apoptotic functions of USP7 inhibitors have also been reported to be mediated by other cellular targets. P5091 and P22077 induced ER/ROS stress and consequently promoted apoptosis in both p53 (+) and (−) HCT116 cells [35]. P5091 also induces apoptosis by inhibiting the deubiquitination of c-Maf and MafB, which are critical transcription factors for myelomagenesis in multiple myeloma cell lines [29]. Treatment with P5901 or depletion of USP7 destabilizes PLK1, the master mitotic regulator and a potential therapeutic target for cancer. It results in cell cycle arrest and apoptosis, which are mitigated with the ectopic expression of PLK1 [44]. P217564, an irreversible inhibitor of the USP7 active site, induces apoptosis in cell lines with both WT and mutant p53 by destabilizing Foxp3/Tip60, ubiquitin-like, with PHD and RING finger domains-1 (UHRF1), and DNA methyltransferase (DNMT1) [28]. HBX19818, a USP7 inhibitor, downregulates homologous recombination repair in chronic lymphocytic leukemia cell lines. Additionally, treatment with HBX19818 promotes DNA damage and consequently promotes DNA damage-induced cell death. HBX19818 upregulates p53 levels without affecting the phosphorylation of p53 [41].

USP7 protects non-malignant cells from apoptosis. P5091 and P22077 induce apoptosis in late-stage erythroblasts and inhibit terminal erythroid differentiation, which requires USP7 activity to remove K48-linked polyubiquitin chains on GATA1 [31]. In thymocytes, caspase-3-dependent processing of USP7 has been reported during apoptosis. This suggests that the USP7/apoptosis axis provides negative feedback for the homeostasis of non-malignant tissues [38]. Future studies must examine the presence of negative feedback mechanisms of USP7 in cell death.

In addition to intrinsic apoptosis, USP7 is involved in extrinsic apoptosis and ferroptosis. USP7 and TRIM27 deubiquitinate RIPK1 during TNFα signaling and promote extrinsic apoptosis. The loss of USP7 or TRIM27 abolishes TNFα/CHX-induced apoptosis in vitro and in vivo [37]. USP7 promotes ferroptosis through p53-dependent modification of chromatin, independent of its transcription factor activity. During erastin-induced ferroptosis, p53 promotes USP7 localization to H2Bub1 of the promoters of SLC7A11 and other iron-binding genes. The complex reduces monoubiquitination of H2B and suppresses the gene expression of ferroptosis-related factors, including SLC7A11 [32].

### 3.3. USP8

USP8, which contains a C-terminal USP domain, is reported to be involved in endosomal trafficking, immune cell regulation, cell death, and Cushing’s disease. Studies on *USP8*-knockout (KO) mice have revealed that USP8 is involved in liver failure and immune cell maintenance [196]. USP8 regulates both intrinsic and extrinsic apoptosis, which are critical for cancer cell survival. Further studies are needed to examine the role of USP8 in other types of RCD.

The depletion of USP8 promotes intrinsic apoptosis by suppressing AKT activity in cholangiocarcinoma (CCA) [79]. USP8 promotes both the stabilization and destabilization of c-FLIP, an anti-apoptotic protein. USP8 protects c-FLIP against proteasomal degradation by cleaving the ubiquitin chain. The depletion of USP8 promotes complex 2 assembly and extrinsic apoptosis by downregulating c-FLIP levels [80]. However, USP8 also promotes c-FLIP degradation in an anti-HER3 antibody treatment model. In this model, USP8 stabilizes ITCH (E3 ligase of c-FLIP), which results in the degradation of c-FLIP and enhanced apoptosis [81].

### 3.4. USP10

USP10 regulates multiple cellular pathways, including DNA damage response, metabolism, ribosome recycling, stress granules, and hematopoiesis [197]. USP10 has also involved in DNA damage-induced apoptosis. Various USP10-dependent molecular cascades suppress apoptosis and paraptosis and consequently modulate human physiology. It includes both malignant and non-malignant tissue homeostasis. Thus, the therapeutic potential of USP10 must be examined by evaluating its role in regulating physiological processes in both diseased and healthy tissues.

USP10 exhibits both anti-apoptotic and proapoptotic functions under DNA damage-induced stress conditions. DNA damage promotes ataxia telangiectasia mutated (ATM)-dependent phosphorylation of USP10, which undergoes nuclear translocation. The DUB activity of USP10 protects p53 from MDM2-mediated degradation and enhances radiation-induced apoptosis mediated by p53 [82]. miRNA-138 negatively regulates the expression of USP10, which results in the destabilization of p53. In contrast, p53 suppresses the transcription of miRNA-138. Thus, miRNA-mediated inhibition of mutant p53 can be a potential therapeutic strategy for malignant tissues [83]. Genotoxic stress promotes the MCPIP1-dependent activation of NF-κB. USP10 cleaves the linear ubiquitin chain from the NF-κB essential modulator (NEMO), resulting in the termination of NF-κB signaling and enhanced apoptosis [84]. Contrary to its role in the apoptotic response against DNA damage, USP10 is directly involved in DNA repair machinery to prevent apoptosis. USP10 deubiquitinates and stabilizes MutS homolog 2 (MSH2), which senses and repairs DNA replication errors. The depletion of USP10 downregulates MSH2 and increases the sensitivity of cells to apoptosis induced by DNA damage-inducing reagents [85].

Additionally, USP10 suppresses other cellular stresses linked to intrinsic apoptosis and facilitates cell survival. The interaction between USP10 and p62 promotes aggresome formation and protects cells from apoptosis induced by the accumulation of ubiquitinated proteins [86]. ROS-induced stress promotes ATM-dependent USP10 activation. Activated USP10 functions as an antioxidant and anti-apoptotic regulator in stress granules. Thus, *USP10*-KO cells are sensitive to arsenite-induced apoptosis [87]. The viral oncoprotein Tax (from human T-cell lymphotropic virus type 1 (HTLV-1)) binds to USP10 and suppresses the anti-apoptotic function of USP10 during ROS stress. This results in increased ROS production and apoptosis in adult T-cell leukemia (ATL), suggesting that USP10 is a potential therapeutic target for ATL [88].

USP10 is also involved in maintaining non-malignant tissue homeostasis and modulation of tissue injury response by suppressing intrinsic apoptosis. In the ischemia-reperfusion model, *USP10*-KO mice exhibited severe injury and inflammation. In this model, USP10 suppresses apoptosis by inhibiting TAK1 [89,90]. USP10 protects endometrial stromal cells from apoptosis and promotes cell proliferation by deubiquitinating Raf-1, which activates the Raf-1/MEK/ERK pathway. The depletion of USP10 upregulates the expression of BAX and caspase-7 and downregulates the expression of BCL-2, which results in enhanced apoptotic cell death [91]. USP10 is involved in hematopoiesis by suppressing apoptosis. *USP10*-KO hematopoietic stem cells exhibit increased apoptotic cell death induced by cytokine deprivation [92].

RNAi screening revealed that USP10 is required for curcumin-induced paraptosis. The knockdown of USP10 or treatment with spoutin-1 (USP10 inhibitor) downregulates mitochondrial dilation, phosphorylation of ERK and JNK, polyubiquitinated protein accumulation, and CHOP [93].

### 3.5. USP11

USP11, which is an X-linked retinal disorder gene, regulates several cellular functions, including histone regulation, NF-κB signaling, DNA repair, and viral infection, and is associated with multiple types of cancers [94]. Additionally, USP11 regulates extrinsic and intrinsic apoptosis and anoikis through various mechanisms. USP11 suppresses apoptosis in cancer cells. In contrast, USP11 promotes cell death in injured tissues. Thus, either by promoting or by suppressing RCD, USP11 promotes pathological processes.

USP11 also regulates DNA damage-induced intrinsic apoptosis. Under genotoxic stress conditions, USP11 stabilizes the p21 protein level, which is upregulated upon DNA damage. The depletion of USP11 inhibits p21 protein expression and increases the sensitivity of cells to apoptosis [94]. USP11 is directly involved in the DNA repair response by deubiquitinating histones, consequently protecting the cells from DNA damage-induced apoptosis [95]. Additionally, USP11 deubiquitinates and stabilizes BRCA2, a DNA repair factor. The knockdown of USP11 increases the sensitivity of cells to mitomycin C [96].

In human breast cancer tissue, USP11 expression was reported to be positively correlated with XIAP. WT USP11 (but not the C318A mutant) deubiquitinates and stabilizes X-linked inhibitor of apoptosis (XIAP) in a xenograft mouse model, which results in the suppression of anoikis and apoptosis and enhanced tumor growth [97].

USP11 deubiquitinates and stabilizes c-IAP-2. The siRNA-mediated knockdown of USP11 or treatment with USP11 inhibitor sensitizes cells to SMAC mimetics and promotes extrinsic apoptosis induced by TRAIL and TNFα [98]. The expression of USP11 is upregulated in an intracerebral hemorrhage model and is correlated with Fas, FasL, and caspase-3. SiRNA-mediated knockdown of USP11 suppressed hemin-induced neuronal cell death. This indicates that USP11 is involved in FasL-mediated apoptosis during brain injury [99].

### 3.6. USP15

USP15, which contains a C-terminal USP domain, is associated with oncogenic signaling, Parkinson’s disease, and antiviral immune response. Additionally, USP15 positively or negatively regulates apoptosis induced by multiple stimuli by targeting various substrates [198]. USP15 modulates cell death through p53, MOMP, NF-κB, and ROS stress. These molecular regulations by USP15 promote or suppress different types of RCD in cancer models. Furthermore, the USP15/RCD axis is involved in anticancer immunity and neuronal cell death. Thus, the USP15/RCD axis is multi-directed and has considerations in its clinical potential.

USP15 promotes p53-mediated apoptosis during high-risk human papillomavirus infection. The interaction between the viral protein E6 and p53 promotes the ubiquitination and degradation of p53. USP15 prevents E6-mediated degradation of p53 and activates apoptosis in the HeLa cell line [100]. The depletion of USP15 decreases the release of cytochrome c into the cytosol during MOMP in extrinsic and intrinsic apoptosis induced by TNFα/actinomycin D or staurosporine, but BAX was not affected. Thus, USP15 promotes MOMP and apoptosis [101]. In chronic myeloid leukemia cell lines, USP15 expression levels were correlated with sensitivity to imatinib-induced apoptosis. STAT5A/miR-202–5p downregulates the expression of USP15 by targeting its mRNA in resistant cells, resulting in the downregulation of caspase-6 [102].

USP15 deubiquitinates and stabilizes MDM2, resulting in suppressing p53 activity and apoptosis in cancer cell lines. Additionally, USP15 promotes nuclear factor of activated T-cells, cytoplasmic 2 (NFATc2) activity, which enhances T-cell activity against tumors. This indicates the complex role of USP15 in tumor progression [103]. In multiple myeloma cell lines, USP15 deubiquitinates and stabilizes NF-κB p65, which results in the inhibition of apoptosis. Moreover, USP15 is a transcriptional target of NF-κB and provides a positive feedforward loop to enhance NF-κB activity and cell survival [104]. In an epilepsy model, siRNA-mediated knockdown of USP15 promoted glutamate-induced apoptosis, which was accompanied by increased oxidative stress, reduced BCL-2, and increased cleaved caspase-3 levels. Thus, USP15 suppresses neuronal death during oxidative damage [105].

### 3.7. USP18

USP18, a representative deISGylating enzyme, reverses the ISG15 conjugation induced by IFN signaling. In addition to ISG-15, USP18 cleaves the ubiquitin chain conjugated to proteins. USP18 is involved in pathogenic infections, inflammation, neurological disorders, and cancer by regulating IFN signaling and other substrates [199]. In this section, we review the USP18/cell death regulatory axis mediated by both deubiquitination and deISGylation. USP18 promotes or suppresses cell death upon activation by IFN signaling. In the absence of an IFN stimulus, USP18 suppresses cell death through alternative molecular cascades. The complex USP18-mediated regulation of cell death has diverse impacts on infectious diseases, tissue degeneration, and cancer.

In IFN-treated acute myeloid leukemia cell lines, depletion of USP18 inhibited cell proliferation and induced apoptosis. *USP18*-KO cells are sensitive to irradiation-induced apoptosis, suggesting that USP18 is a potential therapeutic target [106]. Some studies have revealed the molecular mechanisms involved in the USP18-mediated suppression of apoptosis in IFN stimuli. In type 1 diabetes, USP18 suppresses β-cell inflammation and apoptosis. The depletion of USP18 upregulates the expression levels of IFN target genes, including cytokines, Bim, DP5, and PUMA, which leads to increased mitochondrial death [107]. Additionally, the depletion of USP18 in IFN-treated or bortezomib-treated cancer cells increased extrinsic apoptosis through the upregulation of TRAIL transcription. Deregulated cell death is inhibited upon ectopic expression of FLIP, suggesting that USP18 inhibits IFN-induced cell loss, which is dependent on extrinsic apoptosis [108,109]. However, USP18 promotes apoptosis through IFN signaling in human immunodeficiency virus (HIV) infection. In memory CD4^+^ T cells (Mem) of patients infected with HIV, INF signaling increases the expression level of USP18, which is correlated with increased apoptosis. PTEN expression was upregulated, and pS473 AKT was downregulated, depending on IFN and UPS18. Therefore, USP18 exacerbates HIV infection-related pathology by reducing the maintenance of memory T cells in patients [110].

Furthermore, USP18 suppresses apoptosis independent of IFN signaling by targeting other molecular pathways, including AKT, epidermal growth factor receptor (EGFR), p53, caspases, BCL-2, and Notch1. In cervical cancer cell lines, USP18 suppresses apoptosis through activation of the AKT pathway. Treatment with LY294002, a PI3K/AKT inhibitor, mitigated USP18 overexpression-induced suppression of cell death [111]. The depletion of USP18 increased miR-7 expression, which led to the downregulation of EGFR expression. This negative regulation has been suggested to increase apoptosis in USP18-depleted cells [112]. In hepatitis B virus infection-associated HCC, USP18 suppresses apoptosis by binding and stabilizing BCL-2 [113]. Oxidative stress is reported to promote the expression of USP18, which protects cells against oxidative stress. The depletion of USP18 promotes apoptosis by upregulating the expression of p53 and caspases [114]. Upregulated expression of USP18 in pancreatic cancer suppresses apoptosis by deubiquitinating Notch1, which upregulates c-Myc expression [115].

### 3.8. USP20

USP20 is involved in tumorigenesis, inflammation, metabolism, and DNA repair by regulating diverse cellular substrates [85,200,201,202]. However, studies on USP20 in RCD are limited to intrinsic and extrinsic apoptosis. Interestingly, the regulatory functions of USP20 in RCD are mediated by metabolic regulators. Thus, the clinical implications of the USP20/RCD regulatory axis in pathologies associated with metabolic deregulation must be examined in future studies.

USP20 promotes the TNFα/NF-κB cascade by deubiquitinating and stabilizing p62, which then forms a complex with PKCζ/RIPK1. The depletion of USP20 promotes TNFα/cycloheximide-induced apoptosis [116]. USP20 deubiquitinates and stabilizes ULK1, which is a critical factor for autophagy initiation under starvation conditions. The depletion of USP20 sensitizes cells to starvation-induced apoptosis. Mechanistically, USP20 prevents lysosomal degradation of ULK1 through deubiquitination, which leads to the inhibition of the negative feedback loop and suppression of autophagy [117].

### 3.9. USP22

USP22 regulates various substrates, including histones, telomere repeat binding factor 1 (TRF1), sirtuin 1 (SIRT1), and cyclins. The expression of USP22 is correlated with a poor diagnosis of multiple types of cancers and spinocerebellar ataxia 7 [203]. The oncogenic role of USP22 involves the suppression of apoptosis in cancer cell lines by modulating various substrates. Interestingly, SIRT1 mediates diverse molecular mechanisms involved in the USP22-mediated suppression of intrinsic apoptosis. In addition to SIRT1, USP22 is associated with DNA repair and autophagy in the regulation of RCD. USP22 positively regulates necroptosis in cancer cells. Based on the upregulation of USP22 expression in cancers, the induction of necroptosis can be an effective clinical strategy to kill cancer cells with the active USP22/SIRT1 axis, which inhibits apoptosis.

In gastric cancer, USP22 stabilizes the c-Myc/nicotinamide phosphoribosyltransferase (NAMPT)/SIRT1 signaling cascade, which further activates forkhead box protein O1 (FOXO1) and YAP. The stabilized YAP suppresses apoptosis [118]. USP22 also directly deubiquitinates and stabilizes SIRT1. During embryonic development, stabilized SIRT1 acetylates p53, which suppresses the apoptotic functions of p53. The depletion of USP22 reversed this negative regulation of p53 and enhanced apoptosis in vitro and in vivo [119]. In addition to the post-translational modification (PTM) of p53, overexpression of USP22 downregulates p53, which results in decreased apoptotic cell population in the retinoblastoma cell line [120]. The USP22/SIRT1 signaling axis also activates the AKT/multidrug resistance-associated protein 1 (MRP1) pathway, which promotes resistance to apoptosis in HCC [121]. Furthermore, USP22 enhances resistance to DNA-damaging antitumor drugs directly through the DNA repair mechanism. In the lung adenocarcinoma cell line A549, overexpression of USP22 promotes the repair of cisplatin-induced DNA damage by deubiquitinating H2A. USP22 also stabilizes SIRT1, which then acetylates Ku70, leading to the inhibition of BAX-mediated apoptosis [122]. Moreover, USP22 protects cancer cells by enhancing autophagy. The expression levels of LC3 correlate with the pathological grade of pancreatic cancer. Compared to noncancerous tissue and pancreatic intraepithelial neoplasia (PanIN), the expression levels of LC3 are upregulated in PDAC. Histological analysis of PDAC revealed that the expression pattern of USP22 was similar to that of LC3. USP22 promotes ERK1/2 activity and autophagy in vitro and consequently suppresses apoptotic cell death [123]. Additionally, USP22 protects glioma cell lines from apoptosis through unidentified mechanisms [124]. The pro-tumorigenic functions of USP22 are correlated with the expression of HSP90. The depletion of USP22 sensitizes the cells to HSP90 inhibitors. Thus, a combination therapy targeting USP22 and HSP90 can increase apoptotic cell death in colorectal and breast cancer cells and improve their therapeutic effect [125].

A recent study revealed that USP22 promotes necroptosis. USP22 cleaves the ubiquitin chain from K518 of RIPK3. The loss of USP22 attenuates TNFα/BV6/zVAD-FMK-induced necroptosis but does not affect TNFα/BV6-induced apoptosis and RIPK1 phosphorylation [126].

USP22 also promotes tissue injury-associated cell death. In a diabetes model, USP22 promoted high glucose-induced apoptosis in podocytes by enhancing ROS production, upregulating the expression of caspase-3 and BAX, and promoting the secretion of inflammatory cytokines [127].

### 3.10. CYLD

Cylindromatosis (CYLD) is a tumor suppressor identified in patients with cylindromas in whom both *CYLD* alleles are lost. A similar phenotype of skin tumor development was observed in *CYLD*-KO mice. NF-κB activity and cyclin D1 levels were upregulated in *Cyld*-KO mice. Furthermore, CYLD-deficient cancer cells are resistant to cell death induced by anticancer compounds [204,205,206]. CYLD preferentially cleaves K63- and M1-linked polyubiquitin chains from substrates and cleaves K11- and K48-linked ubiquitin chains [207]. Although CYLD promotes apoptosis and necroptosis by directly regulating their molecular cascades, some studies have indicated indirect regulatory pathways. These findings suggest that CYLD modulates diverse cellular functions, including oncogenic signaling and proteotoxicity, during the regulation of cell death. Thus, other cellular pathways involved in CYLD/RCD regulatory axes must be identified.

CYLD is recruited to the TNFR1 complex, which is dependent on spermatogenesis associated 2 (SPATA2). In the complex, CYLD cleaves K63- and M1-linked polyubiquitin chains. The loss of CYLD stabilizes the TNFR1 complex and NF-κB activity and consequently suppresses TNFα-induced cell death [128,129,130,131,208,209,210]. In TNFR1 signaling, IKK-mediated phosphorylation at serine 418 of CYLD protects cancer cells against cell death by downregulating CYLD activity [211,212,213]. In addition to NF-κB regulation, CYLD promotes cell death by promoting RIPK1 activity. In sharpin-KO mice, the linear ubiquitin chain assembly complex (LUBAC) activity is downregulated. In this model, the downregulated phosphorylation of serine 418 of CYLD enhances the recruitment of RIPK1 to complex 2 by removing the K63-linked polyubiquitin chains of RIPK1, which promotes cell death [132]. During necroptotic cell death, CYLD deubiquitinates RIPK1 and promotes necrosome assembly and activation. Meanwhile, caspase-8, which is activated in complex 2, cleaves CYLD and inhibits necroptosis [133,134,214]. In addition to the PTM of CYLD, miR-454 has been reported to negatively regulate CYLD, promoting cancer cell survival and resistance against oxaliplatin [135]. A recent study reported that CYLD was downregulated in nasopharyngeal carcinoma and that CYLD overexpression promoted apoptosis by upregulating N-myc downstream regulated 1 (NDRG1). This study revealed another tumor-suppressive function of CYLD in inducing cell death, independent of extrinsic death receptor signaling [136].

CYLD is also involved in cardiac injury-induced apoptosis through the regulation of autophagy. A cardiomyopathy study using transverse aortic arch constriction (TAC) revealed that autophagic efflux induced by mTOR reactivation after TAC protects cardiac cells against apoptosis. The overexpression of CYLD in cardiomyocytes results in the accumulation of K48-linked polyubiquitin chains in cells, and the suppression of mTOR activity and autolysosomal clearance leads to cardiac defects and apoptosis [137].

### 3.11. A20

A20 (TNFAIP3) comprises an N-terminal OTU domain with DUB activity and seven C-terminal zinc finger (ZnF) domains. ZnF4 and ZnF7 bind to K63- or M1-linked polyubiquitin chains and exhibit E3 ligase activity. A20, which is reported to be an anti-inflammatory regulator, is involved in multiple regulatory mechanisms in cell death and inflammatory diseases [215]. The precise role of A20 in TNFα-induced cell death depends on the detailed physiological context. In other types of RCD, A20 is a suppressive regulator. Furthermore, the A20/RCD axis is involved in various pathologies, including cancer, tissue injury, pathogen infection, and inflammation. Therefore, A20 is a powerful modulator of various physiologies, coupled with RCD.

Various studies using different models have reported contradictory findings regarding the function of A20 in TNFR1 complex signaling. A20, a pro-survival gene induced by NF-κB, downregulates NF-κB signaling and inhibits sustained inflammation [216]. In addition to the negative feedback regulation of NF-κB signaling, A20 protects cells from TNFα-induced apoptosis. A20 is recruited to the TNFR1 complex, which is dependent on c-IAP-1/2 and inhibits CYLD-mediated degradation of M1-linked ubiquitination of RIPK1, thus enabling the maintenance of pro-survival functions of the TNFR1 signaling pathway. In the absence of M1-linked ubiquitination of RIPK1, A20 suppresses TNFα-induced apoptosis through its DUB activity [138]. However, recent studies have reported that the expression of A20 is upregulated in chronic inflammatory conditions and that A20 promotes TNFα-induced apoptosis by accelerating ripoptosome assembly. The clustered regularly spaced palindromic repeats/caspase-9 (CRISPR-CAS9)-mediated A20-KO in human keratinocytes did not affect TNFα-induced apoptosis, indicating that A20 does not have a pro-survival role in TNFR1 signaling [139].

In contrast, A20 is reported to suppress pathogen-induced apoptosis, DR4/5-induced apoptosis, pyroptosis, and autophagy-related cell death. The depletion of A20 sensitizes immortalized gingival keratinocytes to inflammation and apoptosis induced by *Porphyromonas gingivalis* and TNFα/CHX in a periodontal inflammation model [140]. Treatment with LPS activates microglial cells and promotes the formation of exosomes, which induce neuronal apoptosis. KO of A20 in PC1 cells promotes the apoptotic response to exosomes from microglial cells. This suggests that A20 exhibits anti-apoptotic functions in a brain injury model [141]. In TRAIL/RIPK1-mediated apoptosis, A20 suppresses apoptosis by ubiquitinating RIPK1 with a K63-linked ubiquitin chain at which caspase-8 is recruited and inhibited [142]. In DR4/5-induced apoptosis, A20 hydrolyzes the polyubiquitin chains on caspase-8 generated by cullin-3. This deubiquitination reverses cullin-3-mediated proapoptotic PTM of caspase-8 [143].

A20 regulates pyroptosis by binding to the inflammasome and deubiquitinating caspase-1 at K133 in the presence of RIPK3. The deletion of A20 promotes pyroptotic cell death and inflammation by upregulating caspase-1 activity [144]. In a diabetic nephropathy model, miR-21–5p delivery from high-glucose-stimulated macrophages to podocytes through extracellular vesicles suppresses A20 expression in the podocytes. This results in deregulated pyroptosis and tissue injury due to enhanced inflammasome activity [145].

A20 is also involved in the inflammatory pathology of the intestine. Mass spectrometry analysis revealed that A20 interacted with ATG16L1. Although the ubiquitination of ATG16L1 was not observed in this study, depletion of A20 upregulated the levels of ATG16L1 and LC3-II. The double knockout of A20 and ATG16L1 in the intestinal epithelium results in spontaneous inflammatory bowel disease (IBD)-like pathology. Additionally, the organoid undergoes cell death, which is not rescued upon treatment with the anti-TNFα antibody, zVAD, Nec-1, and zVAD/Nec-1 [146].

### 3.12. OTULIN

OTULIN, which contains an OTU domain, exclusively cleaves M1-linked polyubiquitin chains. Structural studies have revealed that only the M1-linked polyubiquitin chain can exhibit optimal orientation in the catalytic triad of OTULIN [217]. OTULIN is related to inflammatory disease, which is known as OTULIN-related auto-inflammatory syndrome (ORAS). The loss-of-function mutation of OTULIN1 (L272P) results in the accumulation of M1-linked ubiquitin chains, increased NF-κB activity, auto-inflammatory response, and cell death [149,153,218]. Various studies examining the molecular mechanisms regulating RCD and physiological phenotypes have revealed the complex roles of OTULIN in extrinsic cell death. Furthermore, an in vivo study revealed that OTULIN suppresses cell death independent of TNFR1, which has been extensively studied in vitro. Therefore, further studies are needed to elucidate the physiological roles of the OTULIN/RCD axis.

Molecular studies and studies using mouse models have suggested that OTULIN promotes cell death. OTULIN overexpression suppressed LUBAC activity and downregulated the LUBAC-mediated polyubiquitination of complex 1. This results in decreased NF-κB activity, and TNFα signaling shifts to the proapoptotic pathway. Moreover, OTULIN binds to the ubiquitin chain of complex 1 and directly suppresses NF-κB activity, independent of its DUB activity [147,148].

Conversely, OTULIN depletion suppressed LUBAC activity. In these studies, OTULIN-deficient cells were prone to apoptotic or necroptotic cell death. The ubiquitination of TNFR1 complex components is downregulated in mouse embryonic fibroblasts expressing the catalytically inactive OTULIN mutant (C129A). OTULIN prevents the inhibitory self-ubiquitination of LUBAC, which increases the susceptibility of cells to TNFα-induced cell death [149,150,151,152]. Furthermore, OTULIN is regulated during the progression of the cell death cascade to provide additional regulatory steps. Caspase-3-mediated OTULIN cleavage generates a C-terminal fragment, which suppresses apoptosis by limiting caspase activity. OTULIN is phosphorylated during necroptosis, which is inhibited by the phosphatase DUSP14 to suppress cell death [152]. Recently, liver-specific *Otulin*-KO has been shown to promote liver inflammation and damage. Hepatocyte cell death was upregulated in conditional KO mice, which was reversed by *ripk1D138N* knock-in or *Fadd* KO. This indicated that apoptosis mediates the pathology of the *Otulin*-KO liver. Interestingly, the pathological phenotypes in KO mice are not dependent on TNFR but are dependent on mTOR and IFN signaling. Thus, OTULIN may have additional molecular mechanisms to suppress cell death independent of TNFR [153,154].

### 3.13. OTUB1

OTUB1 specifically cleaves the K48-linked ubiquitin chain. OTUB1 also inhibits ubiquitin transfer to the substrate by binding to the E2-Ub complex, independent of DUB activity [219]. OTUB1 promotes or suppresses cell death, depending on the substrate. MDM2/p53 and MDM2/MDMX mediate the proapoptotic function of OTUB1, while other substrates mediate the anti-cell death activities of OTUB1. By regulating various downstream signaling pathways, OTUB1 is involved in intrinsic and extrinsic apoptosis and ferroptosis, which are coupled with cancer and tissue injury.

OTUB1 has been reported to stabilize p53 and consequently mediate DNA damage-induced apoptosis. Mechanistically, OTUB1 downregulated MDM2-mediated ubiquitination of p53. The D88A mutant of OTUB1 could not stabilize p53. Thus, mutant OTUB1 protects p53 through its E2-binding activity [155]. OTUB1 inhibited the anti-apoptotic activity of MDM2 through another substrate (MDMX). OTUB1 cleaves the ubiquitin chain on MDMX conjugated by MDM2, which results in the upregulation of p53 phosphorylation and the activation of apoptotic function [156].

The anti-apoptotic role of OTUB1 has been reported to be mediated through other substrates. FOXM1, a transcription factor related to DNA damage responses, is associated with chemotherapy resistance in cancer. In the epirubicin-resistant cell line, the upregulated expression of OTUB1 stabilizes FOXM1 through deubiquitination. This results in enhanced resistance to chemotherapy/DNA damage-induced apoptosis [157]. A similar suppressive role of OTUB1 in intrinsic apoptosis has also been reported in HCC. OTUB1 expression is correlated with poor clinical outcomes and pathological parameters in patients with HCC. *OTUB1* knockdown in HCC cell lines upregulated proapoptotic BAX and caspase levels and suppressed cell growth [158]. In an intracerebral hemorrhage model (ICH), OTUB1, which was upregulated after ICH, protected against tissue injury. The depletion of OTUB1 upregulates BAX and downregulates BCL-2 protein levels, which results in the induction of hemin-induced apoptosis and tissue injury [159].

In addition to intrinsic apoptosis, OTUB1 also protects cells from extrinsic cell death. In tumor necrosis factor-like weak inducer of apoptosis (TWEAK)-induced apoptosis, OTUB1 stabilizes c-IAP-1 by cleaving its K48-linked ubiquitin chain, which results in the suppression of apoptosis. siRNA-mediated depletion of OTUB1 facilitates apoptosis and decreases NF-κB activation [160].

OTUB1 suppresses ferroptosis by directly stabilizing SLC7A11 protein, in contrast to BAP1 and USP7, which regulate the transcription of SLC7A1. OTUB1 C91A mutant, but not the D88A mutant, could stabilize SLC7A11. This suggests that OTUB1 stabilizes SLC7A11 through its E2-inhibiting activity and not its DUB activity. This positive regulation of SLC7A11 by OTUB1 is enhanced by CD44, which binds to SLC7A11 via a different domain [161].

### 3.14. BAP1

BAP1, a tumor suppressor, is involved in the DNA repair machinery, regulation of cell death, and metabolism. BAP1 mutations result in familial BAP1 cancer syndrome. Various mechanistic studies have revealed the role of BAP1 in multiple cancers [220]. However, BAP1 has also been reported to protect cancer cells against cell death. Thus, the BAP1/RCD axis plays a complex role in tumorigenesis. BAP1 suppresses the expression of pro- or anti-cell death genes depending on the pathological model. Direct protein–protein interactions between BAP1 and substrates also modulate cell death. Some studies have revealed the molecular mechanisms by which cancer cells suppress cell death induced by BAP1 loss. This provided a mechanistic understanding of the survival of malignant cells after BAP1 loss or mutation. Therefore, further studies are needed to identify the role of BAP1 in determining cell fate under diverse pathological conditions (Figure 3b and Figure 4c).

BAP1 is reported to be mutated in malignant mesothelioma (MMe). Cells harboring mutant BAP1 are resistant to gemcitabine, which induces DNA damage and various types of cell death, including apoptosis, necrosis, and necroptosis. The depletion of BAP1 yielded similar results, suggesting that the antitumor function of BAP1 is mediated by the activation of cell death [51]. BAP1 sensitizes cells to ferroptosis by suppressing the expression of SLC7A11, which is dependent on its deubiquitination activity. BAP1 downregulates SLC7A11 by cleaving H2A ubiquitination on the SLC7A11 promoter, which counteracts polycomb repressive complex 1 (PRC1). The expression of BAP1, with mutations associated with human cancers, cannot promote ferroptosis. This suggested that human cancers develop strategies to inhibit ferroptosis by suppressing BAP1 activity [49,54].

BAP1 also induces cell death in cancer cells independent of transcriptional regulation. In the ER, BAP1 colocalizes with and binds to inositol 1,4,5-trisphosphate receptor type 3 (IP3R3). The BAP1-mediated stabilization of IP3R3 promotes Ca^2+^ flux into the mitochondria from the ER, which results in increased mitochondrial apoptosis in cells exposed to DNA-damaging agents [50]. In neuroblastoma cells, BAP1 induces intrinsic apoptosis by releasing BAX from 14–3–3 [47].

However, BAP1 also suppresses apoptosis by regulating gene expression, DNA repair, and ER stress, which overlap with its proapoptotic mechanisms. In cutaneous melanoma tissues, BAP1 is not lost or mutated. Furthermore, shRNA-mediated depletion of BAP1 increases apoptosis and downregulates cell growth. Depletion of BAP1 results in reduced levels of survivin. Thus, BAP1 acts as a tumorigenic protein in this malignancy [46]. BAP1 protects cells from energy-deprivation-induced apoptosis by alleviating ER stress. Mechanistically, BAP1 suppresses the cellular response to energy deprivation by downregulating the expression of the unfolded protein response factor ATF3 and CHOP through its catalytic activity [48]. In malignant pleural mesothelioma, BAP1 and YY1 cooperatively suppress the expression of DR4 and DR5 through the catalytic activity of BAP1. The depletion of BAP1 upregulates the expression levels of DR4 and DR5, which promotes TRAIL-induced apoptosis [45].

Molecular studies have revealed the compensatory mechanisms involved in cancer cells with the loss of BAP1 expression and lack of accelerated cell death. The activity of RNF2 is required to promote apoptosis in BAP1-depleted cells. RNF2 promotes apoptosis by ubiquitinating H2A to silence pro-survival genes (*Bcl2* and *Mcl2*), which is the opposite of the regulation by BAP1. Therefore, cancer cells without RNF2 activity on the promoters of *Bcl2* and *Mcl2* are not sensitized to apoptosis, which facilitates tumorigenesis [53]. In clear cell renal cell carcinoma (ccRCC), the loss or downregulation of BAP1 expression is associated with a poor prognosis. In contrast, the expression of enhancer of zeste homolog 2 (EZH2) is associated with improved prognosis. EZH2 loss or inhibition in BAP1 mutant cells increases apoptosis. This may be due to increased DNA damage because both EZH2 and BAP1 are involved in DNA repair. This implies that cancer cells employ alternative machinery to avoid the cellular dysfunctions of BAP1 loss [52].

### 3.15. UCHL1

The role of UCHL1, which is abundant in the brain, testes, and ovaries, in neuronal diseases has been previously reported [221]. In some cancers, the expression of UCHL1 is silenced through epigenetic regulation. UCHL1 promotes cancer cell apoptosis through diverse molecular pathways. Furthermore, UCHL1 modulates tissue injury and degenerative pathologies by promoting or suppressing cell death. Therefore, UCHL1 is a multifaceted regulator of RCD-related physiology (Figure 4a).

*UCHL1* is frequently methylated in breast and nasopharyngeal carcinoma tissues. Demethylation rescues the expression of UCHL1, which protects p53 against MDM2-mediated degradation. The upregulation of p53 induces apoptosis, which indicates that the tumor-suppressive function of UCHL1 is mediated through the induction of apoptosis [62,67]. In a mouse model of spermatogenesis, UCHL1 promotes developing testes through the induction of apoptosis. The immature testes of *UCHL1*-deficient mice exhibited downregulated levels of apoptotic proteins, such as p53, BAX, and caspase-3 [63,64].

UCHL1 also increases apoptosis in cancer cells through p53-independent mechanisms. In patients with therapy-resistant melanoma and colorectal cancer, UCHL1 is silenced by epigenetic suppression. UCHL1 is reported to be a DUB of NOXA, which promotes genotoxic stress-induced apoptosis. Thus, the loss of UCHL1 expression destabilizes NOXA in chemo-resistant cells and blocks the DNA damage-induced response in resistant cells [65]. In ovarian cancer cell lines, depletion of UCHL1 protects cells from cisplatin-induced apoptosis. The expression levels of anti-apoptotic genes were upregulated, whereas BAX was downregulated upon *UCHL1* knockdown; p53 levels were not altered in this model [69]. In multipotent mesenchymal stromal cells, UCHL1 enhances apoptosis by suppressing BCL-2 levels independent of p53. In this model, the inhibition of UCHL1 suppresses IFNγ- and TNFα-induced apoptosis by upregulating BCL-2 protein levels [57].

UCHL1 is upregulated in diverse tissue injury pathologies and promotes apoptosis. After deep hypothermic circulatory arrest (cardiac surgery), the serum levels of UCHL1 and proapoptotic BAX are significantly correlated with neuronal apoptotic injury [61]. During hypoxia-induced neuronal injury, HIF1α and HIF2α increase the transcription of UCHL1. UCHL1 then mediates apoptotic neuronal cell death and exacerbates hypoxic-ischemic encephalopathy [70]. In a cardiac hypertrophy model (both in vitro and in vivo), UCHL1 expression was upregulated. Downregulation of UCHL1 expression decreases apoptosis and EGFR signaling and suppresses heart remodeling after the onset of cardiac hypertrophy [56]. A recent study demonstrated that podocytes undergo apoptosis and necroptosis during diabetic nephropathy. Diabetic stress upregulates the levels of UCHL1, which deubiquitinates and stabilizes RIPK1 and RIPK3, consequently promoting cell death [59].

Some studies have reported the anti-apoptotic role of UCHL1, especially in tissue injury or degenerative models. UCHL1 relieves the apoptotic burden induced by the accumulation of intracellular toxic oligomers of islet amyloid polypeptide (IAPP) in pancreatic beta (β) cells, resulting in the suppression of type 2 diabetes. The loss of *UCHL1* in human *IAPP* (*hIAPP*) transgenic mice increases the accumulation of lysosomes, IAPP, and cleaved caspase in β cells [68]. During oxygen-glucose deprivation (OGD), a fraction of the cell population exhibited decreased levels of UCHL1. These cells have reduced p27 expression and are prone to ROS-induced apoptosis during OGD and reoxygenation. This indicated that UCHL1 protects cells from OGD-reoxygenation stress [60]. In a mouse model of Alzheimer’s disease, UCHL1 expression was downregulated during ischemic injury. The inhibition of UCHL1 induces apoptosis by upregulating BACE1, whereas restoration of UCHL1 expression protects neuronal cells from apoptosis [55]. UCHL1 prevents ototoxicity, which limits using aminoglycoside antibiotics. RNA sequencing revealed that *UCHL1* is one of the downregulated genes in gentamicin-treated cells. The shRNA-mediated depletion of UCHL1 sensitizes cells to gentamicin-induced cell death and is associated with defects in autophagosomal clearance [58].

UCHL1 was also reported to suppress anchorage-dependent apoptosis (anoikis). The expression of WT, and not that of a catalytically inactive mutant of UCHL1, enhances focal adhesion kinase (FAK)-mediated adhesion signaling, even in the absence of cellular adhesion [66].

### 3.16. UCHL5/USP14

UCHL5, USP14, and RNP11 are the three DUBs in the 19S regulatory particle of the proteasome. DUB activity and allosteric regulation of UCHL5 and USP14 promote or suppress proteasomal degradation of substrates. In various cancers, the expression of UCHL5 and USP14 is upregulated. Several pharmacological inhibitors of UCHL5 and USP14 have been developed [222]. The inhibition of UCHL5 and USP14 induces cell death by regulating multiple substrates and mechanisms (Figure 4b). UCHL5 and USP14 are the shared targets of inhibitors of proteasomal DUB activities, such as b-AP15 and VLX1570. Therefore, in this section, we address the two DUBs.

In drug-resistant cancer cells, inhibitors of UCHL5 and USP14 mitigate resistance and sensitize cancer cells to cell death by modulating diverse cellular mechanisms. These inhibitors induce apoptosis in multiple myeloma by upregulating ER stress [162] and accumulation of proteasome-associated polyubiquitin [163]; in ovarian cancer by suppressing TGF-β signaling [169]; in Waldenström macroglobulinemia by enhancing ER stress and downregulating proteins associated with the B cell receptor [164], regulating genes involved in cellular stress and NF-κB signaling [165], and diffusing large B-cell lymphoma by inhibiting Wnt/β-catenin and TGFβ/Smad pathways [166]; in esophageal squamous cell carcinoma by upregulating the c-Myc/NOXA axis [167]; in leukemia by suppressing aurora kinase B [182]; and in endometrial cancer through an unidentified mechanism [183]. The shRNA-mediated downregulation of UCHL5 suppresses Wnt/β-catenin signaling, which results in enhanced apoptosis in endometrial cancer [170].

Additionally, UCHL5 can induce cell death in a pathogenic infection model. UCHL5 is also involved in *Mycobacterium tuberculosis*-induced macrophage pyroptosis. EST12 secreted by the pathogen cleaves the K48-linked polyubiquitin chains of NLRP3 through UCHL5. Knockdown of UCHL5 completely inhibits EST12-mediated deubiquitination of NLRP3, which suppresses pyroptosis. This indicates that UCHL5 is required for the EST12/RACK/pyroptosis response in macrophages and the clearance of pathogens by the host [171].

To delineate the effect of UCHL5 and USP14, the USP14-specific inhibitor IU1 has been used in diverse pathological models. Depletion or pharmacological inhibition of USP14 revealed the anti-cell death functions of USP14. Treatment with IU1 induces the degradation of COPS6, a negative regulator of p53. Hence, IU1 upregulates p53 and p21 protein levels and consequently promotes apoptotic cell death [172]. In addition, IU1 promotes apoptosis in HeLa cells by destabilizing MDM2 [173]. In breast cancer cells, treatment with IU1 or the shRNA-mediated depletion of USP14 induces K48-linked ubiquitination and degradation of androgen receptors, which promotes apoptotic cell death [174,175]. The depletion of USP14 sensitizes melanoma cells to staurosporine-induced apoptosis, which is independent of mutations in BRAF, NRAS, or TP53. USP14 inhibition results in multiple intracellular stresses, including impaired clearance of polyubiquitinated proteins, increased chaperones, dysfunctional mitochondria, elevated ER stress and increased ROS [176]. The expression of USP14 is upregulated in gastric cancer. The depletion of USP14 sensitizes gastric cancer cells to cisplatin-induced cell death. Mechanistically, the depletion of USP14 downregulates the phosphorylation of AKT/ERK, which is mitigated by treatment with MG132 [177]. A negative cellular regulator of USP14, miR-4782–3p, promotes cell death in non-small-cell lung carcinoma cell lines [178].

Interestingly, UCHL5 and USP14 are associated with the suppression of autophagy, which affects apoptosis and autophagy-dependent cell death. In human podocytes, USP14-mediated deubiquitination and stabilization of SPAG5 contribute to podocyte injury, a clinical predictor of diabetic nephropathy. The USP14/SPAG5-positive-regulatory axis inhibits autophagy and promotes apoptosis by activating AKT/mTOR [179]. The nickel complex NiPT, which is a proteasomal DUB inhibitor, inhibits USP14 and UCHL5 and promotes autophagy associated with the ER stress/AMPK/mTOR/S6K regulatory axis in lung cancer cells. NiPT-induced autophagy mitigated the apoptotic effects of the inhibitor. This suggests that combination with autophagy inhibitors enhances NiPT-induced apoptosis and improves anticancer effects [168]. The expression of USP14 is upregulated in human lung cancer cells. In A549 cells, treatment with USP14-specific inhibitor (IU1-47) or siRNA-mediated USP14 knockdown suppressed tumorigenesis by inducing ER stress-mediated autophagy. USP14 inhibition is mediated by c-Jun N-terminal kinase 1, resulting in autophagic cell death without apoptosis [180,181].

### 3.17. BRCC36

BRCC36 is a component of BRCC, which is a complex comprising BRCA1, BRCA2, and BARD1. Previous studies have reported that BRCC36 is involved in DNA repair [184,185]. BRCC36 suppresses DNA damage-induced apoptosis and promotes pyroptosis. Future studies must focus on elucidating the additional regulatory mechanisms of RCD by BRCC36.

In radiation-induced apoptosis, BRCC36 and BRCC45 enhance the E3 ligase activity of BRCC, promoting the DNA repair activity of BRCC. Thus, BRCC36-mediated suppression of apoptosis is dependent on BRCC. The depletion of BRCC36 impairs BRCA1 phosphorylation and sensitizes cells to ionizing radiation-induced apoptosis [184,185].

BRCC36 also promotes pyroptosis via DUB activity. Treatment with G5, a general DUB inhibitor, suppresses pyroptosis and promotes the ubiquitination of NLRP3. In this study, BRCC36 was identified to be critical for NLRP3 activation by LPS, HLLOMe, and silica [11]. Another study reported that BRCC36 and NLRP3 levels were upregulated during oxidized low-density lipoprotein (oxLDL)-induced pyroptosis. In this model, BRCC36 is required to prevent the degradation of NLRP3 during pyroptosis and to enhance cell death [186]. In the BRCC36/NLRP3 positive regulation axis, the vitamin D receptor (Vdr) is reported to block the interaction between BRCC36 and NLRP3, consequently promoting NLRP3 ubiquitination and mitigating inflammasome-mediated cytokine production and inflammation. *Vdr*-KO mice are highly prone to LPS-induced sepsis, and an additional *NLRP3*-KO rescues the phenotypes [188]. An NLRP3 mutant has been identified in patients with inflammatory bowel disease. The R779C variant of NLRP3, which promotes inflammation, is highly deubiquitinated during inflammasome activation. This deregulated deubiquitination of NLRP3-R779C is mediated by both BRCC36 and JOSD2 [187].

### 3.18. STAMBPL1

STAMBPL1, a member of the AMSH DUB family, is reportedly involved in cancer [191]. Studies on the role of STAMBPL1 in RCD are limited to intrinsic and extrinsic apoptosis. Hence, future studies must examine the additional roles of STAMBPL1 in RCD.

STAMBPL1 directly stabilizes XIAP through deubiquitination and consequently inhibits lysosomal degradation of XIAP. The depletion of STAMBPL1 downregulates XIAP levels and promotes ROS-induced intrinsic apoptosis [189]. The natural compounds honokiol and cepharanthine enhanced TRAIL-induced apoptosis in cancer cell lines. Mechanistically, honokiol and cepharanthine downregulate STAMBPL1, which in turn ubiquitinates and downregulates anti-apoptotic survivin and c-FLIP protein [190,223]. STAMBPL1 expression is upregulated in gastric cancer. Knockdown of STAMBPL1 suppresses malignancy phenotypes, such as cell growth and invasion, and promotes apoptosis [191].

## 4. Conclusions

As summarized in Table 1, various DUBs modulate multiple types of RCD. DUBs directly regulate the components of cell death machinery, such as MOMP, caspases, RIPs, IAPs, NLRP3, and SLC7A11. Furthermore, DUBs can also indirectly regulate cell death by modulating cellular functions by mediating gene expression, DNA repair, ROS-induced stress, clearance of cellular materials, metabolism, and oncogenic signaling.

The diverse molecular mechanisms underlying the modulation of RCD by DUBs lead to the physiological multifunctionality of the DUB/RCD axis. Each DUB/RCD axis modulates several pathological conditions that harbor specific molecular alterations, such as tumorigenesis, tissue injury, degenerative diseases, metabolic disorders, and pathogenic infections. Thus, the regulation of DUB activity is a potential therapeutic strategy for various pathological conditions associated with RCD.

However, the multifunctionality of DUBs and their inhibitors is associated with both advantages and disadvantages for modulating RCD to treat diseases. For example, one DUB may have multiple molecular cascades that converge to similar consequences, such as the suppression of cell death in cancer. Hence, DUB inhibitors induce cancer cell death through multiple mechanisms, which would challenge developing drug resistance in cancer cells. In addition, the inhibitor could be effective against multiple types of cancers with different types of molecular alterations. In other cases, the activities of one DUB may result in either cell death or survival, depending on the cellular context. Hence, inhibitors of DUBs can alleviate or aggravate the disease. Furthermore, different tissues may respond differentially to a single DUB inhibitor, which further complicates the potential application of the DUB/RCD axis as a therapeutic target.

Furthermore, studies on the roles of DUBs in RCD and the associated pathophysiology have been biased toward intrinsic apoptosis, as described in the current review. Therefore, the regulatory function of DUBs and their physiological effects on other types of RCD must be elucidated in future studies.

## Figures and Tables

**Figure 1 ijms-22-04352-f001:**
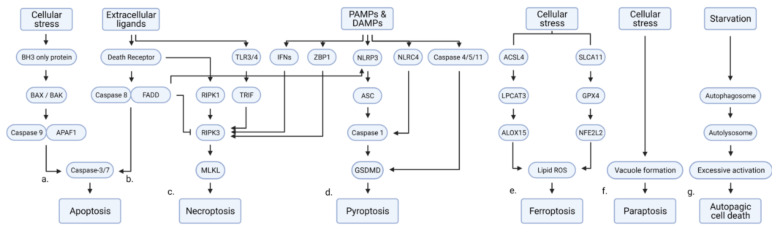
**Molecular cascades of regulated cell death.** (**a**) Intrinsic apoptosis is induced by cellular stresses, such as growth factor depletion, DNA damage, endoplasmic reticulum (ER) stress, reactive oxygen species generation, replication stress, microtubular alterations, and mitotic defects. (**b**) Extrinsic apoptosis is induced by the activation of death receptors by their cognate ligands. Both apoptotic pathways converge at the activation of caspases, which results in cell shrinkage, membrane blebbing, and DNA fragmentation. (**c**) Necroptosis is mediated by activating RIPK3 and MLKL by death receptors, Toll-like receptors, interferon (IFN) signaling, and ZBP1. The sequential phosphorylation of RIPK3 and MLKL is followed by the oligomerization of MLKL, promoting the formation of pores in the plasma membrane. Therefore, necroptosis results in exposure of cellular materials to the surrounding environment. (**d**) Pyroptosis is induced by detecting damage-associated molecular patterns or pathogen-associated molecular patterns, TAK1 inhibition, and activation of death receptors. These stimuli activate gasdermins and promote pyroptotic cell death, which is characterized by pore formation, cell swelling, DNA fragmentation, and chromatin condensation. (**e**) Ferroptosis is induced by impaired regulation of phospholipid hydroperoxides maintained by system Xc−/GPX. Other mediators of ferroptosis are p53, lipid/iron metabolism, and p62. Ferroptosis is associated with the accumulation of iron, lipid peroxidation, and mitochondria shrinkage. (**f**) Paraptosis is accompanied by ER dilation and mitochondrial swelling. The cells exhibit ER/mitochondrial stress and disrupted proteostasis and ion/redox homeostasis. (**g**) Autophagic cell death is initiated by deregulated autophagic activity. Other types of regulated cell death are excluded in its progression. Excessive autophagy of cellular organelles or the activation of Na^+^/K^+^-ATPase results in cell death.

**Figure 2 ijms-22-04352-f002:**
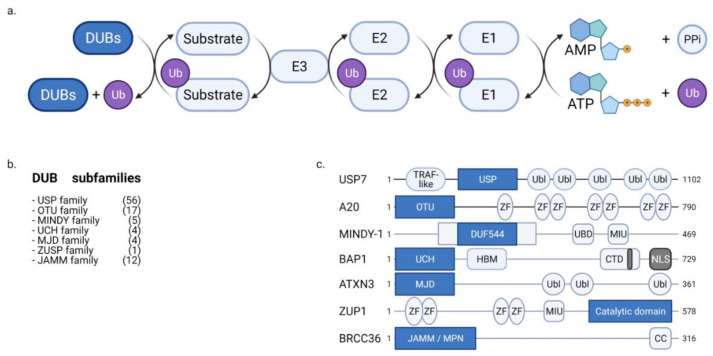
**Ubiquitination/deubiquitination cascades and families of deubiquitinases (DUBs).** (**a**) DUBs cleave ubiquitin conjugated to the substrates. The modulation of ubiquitination status protects the substrates from proteasomal or lysosomal degradation and regulates the signaling capacities of the substrates. (**b**,**c**) DUBs are grouped into USP, OTU, JAMM, MINDY, UCH, MJD, and ZUSP families depending on the characteristic of the conserved domains.

**Figure 3 ijms-22-04352-f003:**
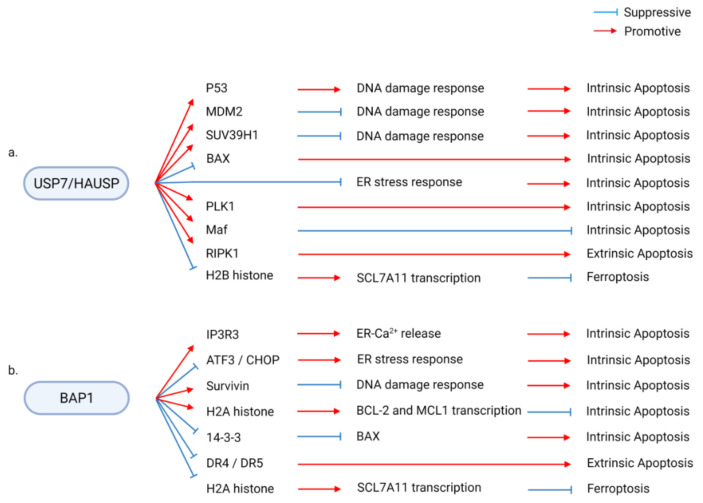
**Examples of diverse molecular pathways involved in the deubiquitinase (DUB)/regulated cell death (RCD) regulatory axis.** DUBs interact with multiple signaling pathways to modulate diverse types of RCD. (**a**) USP7/HAUSP modulates p53, MDM2, SUV39H1, BAX, PLK1, Maf, RIPK1, and H2B histone to enhance or suppress diverse types of RCD, including intrinsic apoptosis, extrinsic apoptosis, and ferroptosis. (**b**) Similarly, the BAP1/RCD axis is mediated by diverse downstream pathways consisting of components, such as IP3R3, ATF3, CHOP, survivin, H2A histone, 14-3-3, and DR4/5.

**Figure 4 ijms-22-04352-f004:**
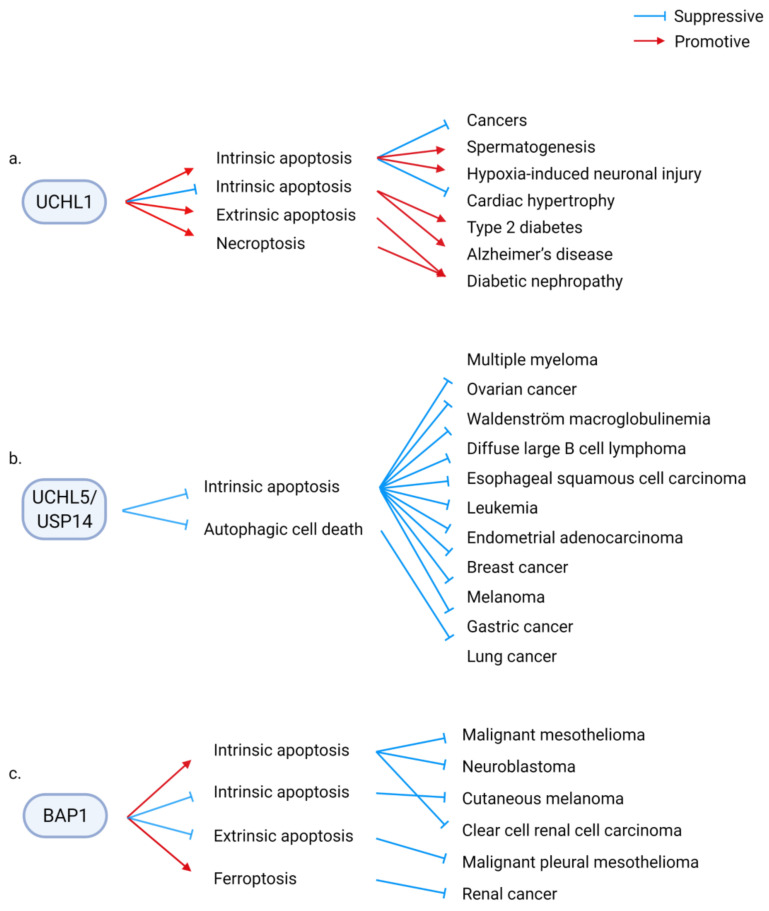
**Examples of diverse pathophysiological consequences of deubiquitinase (DUB)/regulated cell death (RCD) regulatory axes.** DUBs are involved in diverse pathophysiological conditions through the modulation of different types of RCD. (**a**) UCHL1 enhances or suppresses intrinsic apoptosis, extrinsic apoptosis, and necroptosis. The UCHL1/RCD axis aggravates or alleviates malignant diseases, tissue injury, metabolic disorders, and degenerative diseases and is required for spermatogenesis. (**b**) UCHL5/USP14, which are the proteasomal DUBs, aggravate several types of cancer by suppressing cell death. The anticancer potential of the inhibitors of UCHL5 and USP14 has been examined. (**c**) On the contrary, the tumor suppressor, BAP1, promotes or suppresses tumorigenesis by regulating various types of RCD.

**Table 1 ijms-22-04352-t001:** **The list of deubiquitinases regulating multiple types of RCD**.

DUB	Stimulus	RCD	Inhibitory Compound of DUB	Substrate and Molecular Pathway	Associated Physiology and Pathology	Ref.
USP5		↓Intrinsic apoptosis	WP1130	↑c-Maf	↑Multiple myeloma	[71]
		↓Intrinsic apoptosis		↓DNA damage	↑Pancreatic cancer	[72]
	TRAIL, FasL	↓Extrinsic apoptosis	G9(+)	*USP5 is cleaved in TRAIL-sensitive cells	↑Pancreatic cancer, breast cancer	[73]
		↓Intrinsic apoptosis		↓JNK	↑*Drosophila* photoreceptor development	[74]
	*Helicobacter pylori*	↓Intrinsic apoptosis		↓P14ARF, p53	↑Hepatocellular carcinoma	[75,76]
		↓Intrinsic apoptosis		*miR-125a suppresses USP5 expression	↑Multiple myeloma	[77]
	Bortezomib	↓Intrinsic apoptosis		↑hnRNPA1, ↓SF2/ASF1 *SF2/ASF1 suppresses apoptosis in USP5-depleted cells.	↑Glioma	[78]
USP7	Nutlin	↓Intrinsic apoptosis	HBX41108, P5091, C1, C2	↓p53, N-Myc	↑Colorectal cancer	[36]
		↑Intrinsic apoptosis		↑p53	↓Lung cancer	[30]
	Etoposide	↓Intrinsic apoptosis	HBX41108	↑SUV39H1, H3K9me3	↑Colorectal cancer	[40]
	Doxorubicin, etoposide	↓Intrinsic apoptosis	P22077	↓p53	↑Neuroblastoma	[43]
	Doxorubicin	↓Intrinsic apoptosis	P22077	↓BAX	↑Hepatocellular carcinoma	[39]
		↓Intrinsic apoptosis/necrosis	P5091	↑Autophagy	↑Ovarian cancer	[42]
	Senescence	↓Intrinsic apoptosis	P5091/P22077	↓p53	↑Senescence-associated secretory phenotype	[33]
		↓Intrinsic apoptosis	P5091/P22077	↓ER/ROS stress	↑Colorectal cancer	[35]
		↓Intrinsic apoptosis	P5091	↑c-Maf, MafB	↑Multiple myeloma	[29]
	Paclitaxel, docetaxel	↓Intrinsic apoptosis	P5091	↓PLK1	↑Prostate cancer	[44]
		↓Intrinsic apoptosis	P217564	↑Foxp3, Tip60, UHRF1, DNMT1	↑Colorectal cancer, prostate cancer, leukemia	[28]
	Irradiation	↓Intrinsic apoptosis	HBX19818	↑Homologous recombination repair	↑Leukemia	[41]
		↓Intrinsic apoptosis	P5091/P22077	↑GATA1	↑Erythroblast differentiation	[31]
	Dexamethasone, irradiation			* Activated caspase-3 cleaves USP7	Thymocyte development	[38]
USP7	TNFα, cycloheximide	↑Extrinsic apoptosis		↑RIPK1	↑Liver injury	[37]
	Erastin	↑Ferroptosis		↓H2Bub1	↓Lung cancer	[32]
USP8		↓Intrinsic apoptosis		↑AKT	↑Cholangiocarcinoma	[79]
	αFas, TRAIL, cycloheximide	↓Extrinsic apoptosis		↑c-FLIP	↑Cervical cancer, melanoma	[80]
	9F7-F11	↑Extrinsic apoptosis		↓ITCH, c-FLIP	↓Pancreatic cancer, breast cancer, prostate cancer	[81]
USP10	Irradiation	↑Intrinsic apoptosis		↑p53	↓Colorectal cancer, prostate cancer	[82]
		↑Intrinsic apoptosis		↑p53 *miR-138 suppresses USP10 expression	↓Colorectal cancer, cervical cancer, lung carcinoma, non-small cell lung cancer cell line	[83]
	Etoposide, camptothecin, irradiation	↑Intrinsic apoptosis		↓NEMO	↓Fibrosarcoma	[84]
	MNNG, 6-TG	↓Intrinsic apoptosis		↑MSH2	↑Lung cancer	[85]
	MG-132, bortezomib	↓Intrinsic apoptosis		↑p62	↓Parkinson’s disease	[86]
	Arsenite	↓Intrinsic apoptosis		↓ROS-induced stress	↑Osteosarcoma, adrenal cortex adenocarcinoma, cervical cancer	[87]
	Arsenite	↓Intrinsic apoptosis		↓ROS-induced stress *Tax suppresses the activity of USP10	↑Adult T-cell leukemia	[88]
	Ischemic injury	↓Intrinsic apoptosis		↓TAK1	↓Ischemia-reperfusion	[89,90]
		↓Intrinsic apoptosis		↑Raf-1	↑Endometriosis	[91]
	Cytokine deprivation	↓Intrinsic apoptosis			↑Hematopoiesis	[92]
	Curcumin	↑Paraptosis	Spoutin-1	↑ERK, JNK, ER stress, mitochondrial dilation	↓Breast cancer	[93]
USP11	Etoposide, doxorubicin	↓Intrinsic apoptosis		↑p21	↑Lung cancer, colorectal cancer	[94]
	Camptothecin, VP-16	↓Intrinsic apoptosis		↓Histone ubiquitination	↑Cervical cancer, osteosarcoma	[95]
	Mitomycin C	↓Intrinsic apoptosis		↑BRCA2	↑Breast cancer, pancreatic cancer	[96]
	Loss of adhesion, cisplatin	↓Intrinsic apoptosis, anoikis	Mitoxantrone	↑XIAP	↑Breast cancer	[97]
	TRAIL, TNFα, SMAC mimetics	↓Intrinsic apoptosis	Mitoxantrone	↑c-IAP-2	↑Colorectal cancer, melanoma	[98]
	Hemin	↑Extrinsic apoptosis		↑Fas, FasL, caspase-3	↑Intracerebral hemorrhage	[99]
USP15	Papilloma virus	↑Intrinsic apoptosis		↑p53	↓Papilloma virus infection	[100]
	TNFα, actinomycin D, staurosporine	↑Intrinsic, extrinsic apoptosis		↑Cytochrome c release	↑Breast cancer, cervical cancer	[101]
	Imatinib	↑intrinsic apoptosis		↑Caspase-6 *SPATA5A/miR-202-5p suppresses USP15 expression	↓Chronic myeloid leukemia	[102]
		↓Intrinsic apoptosis		↑MDM2 *USP15 promotes T-cell activity against cancer cell	↑Colorectal cancer, melanoma	[103]
		↓Intrinsic apoptosis		↑p65	↑Multiple myeloma	[104]
	Glutamate induced oxidative stress	↓Intrinsic apoptosis		↑ROS, caspase, ↓Bcl-2	↓Epilepsy	[105]
USP18	IFN, irradiation	↓Intrinsic apoptosis			↑Acute myeloid leukemia	[106]
	IFN	↓Intrinsic apoptosis		↓STAT/DP5, PUMA, Bim	↓Type 1 diabetes	[107]
	IFN, bortezomib	↓Extrinsic apoptosis		↓TRAIL	↓Breast cancer	[108,109]
	HIV, IFN	↑Intrinsic apoptosis		↑PTEN, ↓AKT	↑HIV infection	[110]
		↓Intrinsic apoptosis		↓AKT	↑Cervical cancer	[111]
		↓Intrinsic apoptosis		↓miR-7, ↑EGFR	↑Glioma, cervical cancer	[112]
		↓Intrinsic apoptosis		↑Bcl-2	↑HBV-associated hepatocellular carcinoma	[113]
	Oxidative stress	↓Intrinsic apoptosis		↓p53, caspase	↓Oxidative liver injury	[114]
		↓Intrinsic apoptosis		↑Notch1, c-MYC	↑Pancreatic cancer	[115]
USP20	TNFα, cycloheximide	↓Extrinsic apoptosis		↑p62	↑Cervical cancer	[116]
	HBSS	↓Intrinsic apoptosis		↑ULK1	↑Cervical cancer	[117]
USP22		↓Intrinsic apoptosis		↑c-MYC	↑Gastric cancer	[118]
	Etoposide	↓Intrinsic apoptosis		↑SIRT1, ↓p53	↑Embryonic development	[119]
		↓Intrinsic apoptosis		↓TERT, p53	↑Retinoblastoma	[120]
	5-FU	↓Intrinsic apoptosis		↑SIRT1/AKT/MRP1	↑Hepatocellular carcinoma	[121]
	Cisplatin	↓Intrinsic apoptosis		↓H2A ubiquitination, ↑SIRT1	↑Lung adenocarcinoma	[122]
		↓Intrinsic apoptosis		↑ERK1/2, Autophagy	↑Pancreatic cancer	[123]
		↓Intrinsic apoptosis			↑Glioma	[124]
	Ganetespib	↓Intrinsic apoptosis		↑HSP90	↑Colorectal cancer, breast cancer	[125]
	TNFα, BV6, zVAD-FMK	↑Necroptosis		↑RIPK3	↓Colorectal cancer	[126]
	High glucose stress	↑Intrinsic apoptosis		↑Caspase-3, Bax, inflammation	↑Diabetic nephropathy	[127]
CYLD	TNFα, cycloheximide	↑Extrinsic apoptosis		↓TNFR/NF-κB		[128,129,130,131]
	TNFα, cycloheximide, zVAD-FMK	↑Extrinsic apoptosis, necroptosis		↑RIPK1	↑Autoinflammation, immunodeficiency	[132]
	TNFα, LPS, LBW242, zVAD-FMK	↑Necroptosis		↑RIPK1/necrosome *Caspase-8 cleaves CYLD		[133,134]
	Oxaliplatin	↑Intrinsic apoptosis		* miR-454 suppresses CYLD	↓Gastric cancer	[135]
		↑Intrinsic apoptosis		↑NDRG1	↓Nasopharyngeal carcinoma	[136]
	Transverse aortic arch constriction	↑Intrinsic apoptosis		↓mTOR/autolysosomal clearance	↑Cardiomyopathy	[137]
A20	TNFα	↓Extrinsic apoptosis		↓TNFR/NF-κB		[138]
	TNFα	↑Extrinsic apoptosis		↑TNFR/NF-κB/NIK/ripoptosome	↑Psoriasis	[139]
	*Porphyromonas gingivalis*, TNFα, cycloheximide	↓Intrinsic, extrinsic apoptosis		↓Caspase-3	↓Periodontal disease	[140]
	LPS-induced microglial exosome	↓Intrinsic apoptosis		↓Caspase-3, Bax, ↑Bcl-2	↓Brain injury	[141]
	TRAIL	↓Extrinsic apoptosis		↓RIPK1/caspase-8	↑Glioblastoma	[142]
	TRAIL	↓Extrinsic apoptosis		↓Caspase-8	↑Lung cancer, colorectal cancer	[143]
		↓Pyroptosis		↓Caspase-1		[144]
	High glucose	↓Pyroptosis		↓NLRP3 *miR-21-5p suppresses A20	↑Diabetic nephropathy	[145]
		↓Unclassified cell death		↓ATGL16L1	↓IBD-like pathology	[146]
OTULIN	TNFα, L18-MDP	↑Extrinsic apoptosis		↓LUBAC, NF-κB	↓Cervical cancer, osteosarcoma	[147,148]
	TNFα, cycloheximide, zVAD-FMK	↓Extrinsic apoptosis, necroptosis		↑LUBAC, ↓caspase *DUSP14 dephosphorylates OTULIN	↓OTULIN-related autoinflammatory syndrome (ORAS)	[149,150,151,152]
		↓Extrinsic apoptosis		↓mTOR, IFN	↓ORAS-associated liver diseases, hepatocellular carcinoma	[153,154]
OTUB1	Neocarzinostatin, etoposide, UV	↑Intrinsic apoptosis		↑p53	↓Osteosarcoma	[155]
	UV	↑Intrinsic apoptosis		↑MDMX	↓Osteosarcoma	[156]
	Epirubicin	↓Intrinsic apoptosis		↑FOXM1	↑Breast cancer	[157]
		↓Intrinsic apoptosis		↓Bax, caspase	↑Hepatocellular carcinoma	[158]
	Intracerebral hemorrhage, hemin	↓Intrinsic apoptosis		↓Bax, ↑Bcl-2	↓Brain injury	[159]
	TWEAK, BV6	↓Extrinsic apoptosis		↑c-IAP-1	↑Ovarian carcinoma, melanoma, fibrosarcoma, breast cancer	[160]
	tert-butyl hydroperoxide	↓Ferroptosis		↑SLC7A11	↑Non-small cell lung cancer, neuroblastoma, osteosarcoma	[161]
BAP1	Gemcitabine	↑Intrinsic apoptosis			↓Malignant mesothelioma	[51]
	Erastin	↑Ferroptosis		↓H2A ubiquitination	↓Renal cell carcinoma	[49,54]
	H2O2, C2-ceramide, menadione, 5-FU, irradiation, UV	↑Intrinsic apoptosis		↑IP3R3	↓Pleural mesothelioma,	[45]
		↑Intrinsic apoptosis		↑Bax	↓Neuroblastoma	[47]
	Reduced serum	↓Intrinsic apoptosis		↑Survivin	↑Cutaneous melanoma	[46]
	Glucose deprivation	↓Intrinsic apoptosis		↓ATF3, CHOP	↑Renal cell carcinoma	[48]
	TRAIL	↓Extrinsic apoptosis		↓DR4/5	↑Pleural mesothelioma	[45]
		↓Intrinsic apoptosis		↓Histone ubiquitination of promoters of *Bcl2* and *Mcl2*	↑Survival of RNF2-active cancer	[53]
BAP1		↓Intrinsic apoptosis		↓DNA repair *EZH2 protects cancer cells without active BAP1	↑Clear cell renal cell carcinoma	[52]
UCHL1		↑Intrinsic apoptosis		↑p53	↓Nasopharyngeal carcinoma, breast cancer	[62,67]
		↑Intrinsic apoptosis		↑p53, Bax, caspase-3	↑Spermatogenesis	[63,64]
	Doxorubicin, cisplatin, irradiation	↑Intrinsic apoptosis	LDN-57444	↑NOXA	↓Melanoma, colorectal cancer	[65]
UCHL1	Cisplatin	↑Intrinsic apoptosis		↑Bax, ↓BCL2, BCL11A, AEN, XIAP, AKT	↓Ovarian cancer	[69]
	IFN, TNFα	↑Extrinsic apoptosis	LDN-57444	↓Bcl-2	↓Multipotent mesenchymal stromal cell survival	[57]
	Deep hypothermic circulatory arrest	↑Intrinsic apoptosis			↑Neuronal injury by cardiac dysfunction	[61]
	Hypoxia	↑Intrinsic apoptosis			↑Hypoxic-ischemic encephalopathy	[70]
	Transverse aortic constriction	↑Intrinsic apoptosis	LDN-57444	↑EGFR	↓Cardiac hypertrophy	[56]
	High glucose	↑Apoptosis, necroptosis		↑RIPK1, RIPK3	↑Diabetic nephropathy	[59]
	IAPP accumulation	↓Intrinsic apoptosis		↑Lysosomal degradation	↓Type 2 diabetes	[68]
	Oxygen-glucose deprivation	↓Intrinsic apoptosis		↓p27	↓Embryonal carcinoma, neuroblastoma	[60]
	Ischemic injury	↓Intrinsic apoptosis		↓BACE1	↓Alzheimer’s disease	[55]
	Gentamicin	↓Intrinsic apoptosis		↑Autophagosomal clearance	↓Ototoxicity of antibiotics	[58]
	Loss of adhesion	↓Anoikis		↑FAK	↑Cervical cancer	[66]
UCHL5/ USP14	SAHA, lenalidomide, deexamethasone	↓Intrinsic, extrinsic apoptosis	b-AP15	↓ER stress	↑Multiple myeloma	[162]
		↓Intrinsic apoptosis	b-AP15, VLX1570	↓Proteasome-associated polyubiquitin	↑Multiple myeloma	[163]
	Ibrutinib	↓Intrinsic apoptosis	b-AP15, VLX1570	↑NF-κB, BCR-associated genes, ↓Cellular stress	↑Waldenström macroglobulinemia	[164,165]
		↓Intrinsic apoptosis	b-AP15	↑Wnt, TGFβ	↑Diffuse large B-cell lymphoma	[166]
		↓Intrinsic apoptosis	b-AP15	↑c-MYC/NOXA	↑Esophageal squamous cell carcinoma	[167]
		↓Intrinsic apoptosis	NiPT		↑Lung cancer	[168]
UCHL5		↓Intrinsic apoptosis	b-AP15	↑TGFβ	↑Ovarian cancer	[169]
		↓Intrinsic apoptosis		↑Wnt	↑Endometrial cancer	[170]
	*Mycobacterium tuberculosis*	↑Pyroptosis		↑NLRP3	↓*Mycobacterium tuberculosis* infection	[171]
USP14		↓Intrinsic apoptosis	IU1	↑COPS6	↑Osteosarcoma, B cell lymphoma	[172]
		↓Intrinsic apoptosis	IU1	↓MDM2	↑Cervical cancer	[173]
	Enzalutamide	↓Intrinsic apoptosis	IU1	↑Androgen receptor	↑Breast cancer	[174,175]
	Staurosporine	↓Intrinsic apoptosis		↑Mitochondria homeostasis ↓ER stress, ROS	↑Melanoma	[176]
	Cisplatin	↓Intrinsic apoptosis		↑AKT/ERK	↑Gastric cancer	[177]
		↓Intrinsic apoptosis		*miR-4782-3p suppresses USP14	↑Non-small cell lung cancer	[178]
	High glucose	↑Intrinsic apoptosis		↑SPAG5	↑Diabetic nephropathy	[179]
		↓Autophagic cell death	IU1	↓ER stress-mediated autophagy, JNK	↑Lung cancer	[180,181]
		↓Intrinsic apoptosis	b-AP15	↑Aurora B	↑Leukemia	[182]
		↓Intrinsic apoptosis	VLX1570		↑Endometrial cancer	[183]
BRCC36	Irradiation	↓Intrinsic apoptosis		↑BRCC	↑Breast cancer, cervical cancer	[184,185]
	LPS, HLLOMe, silica, oxLDL	↑Pyroptosis	G5	↑NLRP3	↑Pyrogenic response	[11,186,187]
	LPS	↑Pyroptosis		↑NLRP3 *Vdr suppresses BRCC36	↑Pyrogenic response	[188]
STAMBPL1		↓Intrinsic apoptosis		↑XIAP	↑Prostate cancer	[189]
	TRAIL	↓Extrinsic apoptosis	Honokiol, cepharanthine	↑Survivin, c-FLIP	↑Renal cell carcinoma, lung cancer, kidney carcinoma	[189,190]
		↓Intrinsic apoptosis		↓NF-κB	↑Gastric cancer	[191]

*: Upstream regulators of *DUB*.

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
