# Peer review of "Deubiquitinases: Modulators of Different Types of Regulated Cell Death"

_ijms, 2021, doi:10.3390/ijms22094352_

Round 1

Reviewer 1 Report

Lee et al. review diverse mechanisms by which deubiquitinases (DUBs) control different types of regulated cell death (RCD). It is impressive that the authors summarize a vast amount of literature in the field by describing the roles of 18  DUBs in RCD. Differential mechanistic contributions by multiple DUBs should provide the readers with insights on how we treat associated pathologies. However, it seems that the current manuscript lacks coherence for delivering key points, which should have been something like 'how each type of RCD is regulated by which DUBs' and 'successes and challenges to development of therapeutic strategies based on knowledge of DUBs in RCDs', for instance.

The authors list up findings in the field about 18 DUBs in 17 subsections and suddenly reach to the conclusion with less correlation with descriptions in the 17 subsections. Expectation went up as the reviewer read the title and the abstract, but down as the long list of individual DUBs appeared throughout the manuscript.

Hints on reorganization of the manuscript can be found in Figures 3 and 4 and 'Conclusion' section. First of all, the authors did not refer to either Figure in the main text, apparently. Rather than simple listing up individual DUBs, the authors are encouraged to divide the main session into two topics: 1) DUBs and their mechanisms according to types of RCD. There should be "take-home points" even in a review paper; 2) Diverse pathophysiological consequences of DUB/RCD regulatory axes.

Contents in the 17 subsections under the title '2. DUBs regulating diverse RCDs' can be summarized in a Table. In such a table, the following entries can be described, for example: DUB, RCD associated, mechanisms/signaling axis or pathway, and pathologies associated.

Author Response

•First of all, the authors did not refer to either Figure in the main text, apparently
- We added additional parts describing the figures and referred all the figures as the reviewer suggested.

•Rather than simple listing up individual DUBs, the authors are encouraged to divide the main session into two topics: 1) DUBs and their mechanisms according to types of RCD. There should be "take-home points" even in a review paper; 2) Diverse pathophysiological consequences of DUB/RCD regulatory axes. Contents in the 17 subsections under the title '2. DUBs regulating diverse RCDs' can be summarized in a Table. In such a table, the following entries can be described, for example: DUB, RCD associated, mechanisms/signaling axis or pathway, and pathologies associated.
Contents in the 17 subsections under the title '2. DUBs regulating diverse RCDs' can be summarized in a Table. In such a table, the following entries can be described, for example: DUB, RCD associated, mechanisms/signaling axis or pathway, and pathologies associated.
-We agree with reviewer’s suggestion to rearrange the manuscript in tow topics. However, the role of each DUB is diverse and involved in the multiple RCDs at the same time. We tried to describe how the downstream pathways of each DUB regulating RCD are associated with completely different outcomes. These make it extremely hard to categorize this manuscript into the two parts. To make the manuscript more efficient and concise, we included the table summarizing all the DUBs in the manuscript as the reviewer suggested. The table is rather comprehensive since it contains information on the stimuli of RCD, type of RCD, molecular pathways, chemical inhibitors of DUBs, and diseases related, which we expect to help reader better understand the manuscript. 

Reviewer 2 Report

The review discusses the different mechanisms involved in cell death.
The paper is wide, well argued, convincing, provides good scientific literature, it is still generic in the conclusions, perspectives.

The figures appear too small. I would recommend to enhance them.

In addition, a section of abbreviations should be inserted.

Author Response

•The figures appear too small. I would recommend to enhance them.
-We enhanced the figures as suggested.

•In addition, a section of abbreviations should be inserted.
-We included the abbreviation as suggested.

Round 2

Reviewer 1 Report

The authors answered the issues raised to some extents.

The Supplementary Table 1 in the revised manuscript would be better suited to be a regular Table 1 in the main text, given that there is no apparent limit on the page numbers set by the journal and that most readers would seek tables or figures first.

In the Introduction section, the authors may want to very briefly touch that atypical types of ubiquitination such as that occurring on Serine and Cysteine residues (1 or 2 sentences would be fine).

Author Response

Dear reviewer, we edited the manuscript as suggested in 2nd round of revision.
We got one more English editing service for minor spell check. 
And we manipulated some parts of manuscript as described bellow.

1. The Supplementary Table 1 in the revised manuscript would be better suited to be a regular Table 1 in the main text, given that there is no apparent limit on the page numbers set by the journal and that most readers would seek tables or figures first.
- We inserted Table 1 in the main text following reviewer's suggestion. (page 8, line number : 285)
-Sentences mentioning the tables are edited. (Supplementary Table 1 -> Table 1) (page 8, line number : 282 , page 26, line number : 1000)

2. In the Introduction section, the authors may want to very briefly touch that atypical types of ubiquitination such as that occurring on Serine and Cysteine residues (1 or 2 sentences would be fine).
-We modified some sentences to briefly handle atypical types of ubiquitination following reviewer’s suggestion. (Page 5, line number 194~205)

Round 3

Reviewer 1 Report

The authors answered the concerns satisfactorily. One last fine thing to ask would be as follows:

Line 194: canonical ubiquitination typically implies involvement of lysines and methionine-1 (M1). Methionine is not mentioned here.